# A defensin-like protein drives cadmium efflux and allocation in rice

Jin-Song Luo[1,2,3], Jing Huang[1,4], Da-Li Zeng[5], Jia-Shi Peng[1], Guo-Bin Zhang[1], Hai-Ling Ma[1,2], Yuan Guan[1], Hong-Ying Yi[1], Yan-Lei Fu[1], Bin Han[1], Hong-Xuan Lin[1], Qian Qian[5] & Ji-Ming Gong [1,2]

Pollution by heavy metals limits the area of land available for cultivation of food crops. A potential solution to this problem might lie in the molecular breeding of food crops for phytoremediation that accumulate toxic metals in straw while producing safe and nutritious grains. Here, we identify a rice quantitative trait locus we name cadmium (Cd) accumulation in leaf 1 (CAL1), which encodes a defensin-like protein. CAL1 is expressed preferentially in root exodermis and xylem parenchyma cells. We provide evidence that CAL1 acts by chelating Cd in the cytosol and facilitating Cd secretion to extracellular spaces, hence lowering cytosolic Cd concentration while driving long-distance Cd transport via xylem vessels. CAL1 does not appear to affect Cd accumulation in rice grains or the accumulation of other essential metals, thus providing an efficient molecular tool to breed dual-function rice varieties that produce safe grains while remediating paddy soils.

[1] National Key Laboratory of Plant Molecular Genetics and CAS center for excellence in Molecular Plant Sciences, Shanghai Institute of Plant Physiology and Ecology, Chinese Academy of Sciences, Shanghai, 200032, China. [2] University of Chinese Academy of Sciences, Beijing, 100049, China. [3] Southern Regional Collaborative Innovation Center for Grain and Oil Crops in China, Hunan Agricultural University, Changsha, 410128, China. [4] State Key Laboratory for Conservation and Utilization of Subtropical Agro-bioresources, College of Life Science and Technology, Guangxi University, Nanning, 530004, China. [5] China National Rice Research Institute, Hangzhou, 310006, China. Jin-Song Luo, Jing Huang and Da-Li Zeng authors contributed equally to this work. Correspondence and requests for materials should be addressed to Q.Q. (email: qianqian188@hotmail.com) or to J.-M.G. (email: jmgong@sibs.ac.cn)

Heavy metal pollution is becoming a worldwide concern to both agriculture and human health[1–4]. In China alone, nearly 20 million hectares of the cultivated land is contaminated with cadmium (Cd), arsenic (As), and/or lead (Pb), resulting in about 12 million tons of contaminated grains which translate to about 20 billion Yuan/2.4 billion US dollars of economic loss per year[5]. Rice is an important staple food for people especially in East or Southeast Asia, where economy grows very fast and heavy metal contamination is widespread[6, 7]. Cd concentration in rice was found as high as $1-2\,mg\,kg^{-1}$ in some areas[8], much higher than China's national standard $0.2\,mg\,kg^{-1}$ (GB2762-2005) and the provisionally safe threshold $0.4\,mg\,kg^{-1}$ as defined by the Codex Alimentarius Commission of Food and Agriculture Organization/World Health Organization[9], imposing great threat to both food security and food safety. Therefore, remediation of paddy soils and reducing heavy metal accumulation in rice grains become two urgent issues to be tackled, while both require better understanding of heavy metal accumulation and translocation in rice plants.

Defensins are a group of small cysteine rich proteins, sharing the motif CSαβ that is composed of a α-helix and a triple-stranded anti-parallel β-sheet[10]. They display a globular three-dimensional structure which is believed to be stabilized by four disulfide bridges[11, 12]. Defensins are widely distributed in living species (invertebrates, plants, and mammals) and known for their involvement in the ancestral non-specific innate immune response and antifungal activity[13, 14]. Their possible role in metal physiology was not established until most recently when one study reported Zn/Cd binding activity of the human defensin 5 (HD5) while others indicated that plant defensin enhanced Zn tolerance[15–17], however, the underlying working machinery remains largely unknown.

Here, we report the identification of CAL1, which encodes a defensin-like protein, that positively regulates Cd accumulation in rice leaves. We provide evidence that CAL1 acts via chelation of Cd and secretion to extracellular space thus enhancing Cd allocation in rice.

## Results

**Map based cloning of CAL1.** To elucidate the mechanisms of Cd accumulation and translocation in rice, we performed an ionomic screening[18] of 212 rice accessions, which include representative rice cultivars and a mini core collection that is believed to cover >70% of the genetic diversity[19]. Our screening identified one *indicia* cultivar Tainan1 (TN1) over-accumulating Cd (Supplementary Fig. 1, Supplementary Table 1). Cd levels in TN1 grains (Fig. 1a) and leaves (Fig. 1b) were 3–4-fold of those in the under-accumulating cultivar Chunjiang06 (CJ06), while no difference was observed for iron (Fe), zinc (Zn), manganese (Mn), and copper (Cu) (Fig. 1a, b). Further analysis showed that Cd concentration increased more significantly in TN1 leaf (Fig. 1c) and xylem sap (Fig. 1d) compared to CJ06 under different Cd treatments, while apparent difference was not detected in Cd uptake (Supplementary Fig 2a, b) or metal tolerance (Supplementary Fig. 2c, d) between TN1 and CJ06, suggesting that increased translocation might represent a major contribution to the higher Cd accumulation in TN1 aerial parts. Using a double haploid (DH) population derived from TN1 and CJ06[20], we first mapped quantitative trait loci (QTLs) controlling Cd accumulation in leaves, and 3 QTLs were identified (Fig. 2e and Supplementary Table 2), among which a QTL on chromosome 2 explained ~13% of the variance detected (Supplementary Table 2). It was selected for further characterization and assigned the name *CAL1* (Cd Accumulation in Leaf 1).

To clone the causal gene for *CAL1*, the Cd over-accumulation cultivar TN1 was repetitively backcrossed to CJ06, and a chromosome segment substitution line (CSSL) HF8 was obtained using marker-assisted selection, which carried a segment of *CAL1* from TN1 and showed increased Cd concentration in xylem sap (Fig. 1f). HF8 was further backcrossed to CJ06 and high-resolution mapping was performed. Using 3651 $BC_3F_3$ progenies, *CAL1* was localized to a 56 kb genomic region, where four putative genes encoding an F-box family member (Os02g0630000), an auxin responsive factor (Os02g0628600), an aquaporin (Os02g0629200) and a defensin family member (Os02g0629800), respectively, were predicted (Fig. 1g and Supplementary Table 3). Comparative sequencing showed that several SNPs occurred in exons of Os02g0630000, Os02g0628600 and Os02g0629200 between the parental alleles, respectively, leading to transitions in amino acid sequences (Supplementary Fig. 3a). In Os02g0629800, however, nucleotide polymorphism was only observed in non-coding sequences (one SNP in 5′-UTR, 21 in the promoter region, and a 36bp-InDel in the first intron, Fig. 1g and Supplementary Fig. 3b). Complementation assay using full genomic sequence of each gene showed that only Os02g0629800 increased Cd accumulation in CJ06 leaves (Supplementary Fig. 4a–h). Therefore, we proposed that Os02g0629800 was the causal gene for *CAL1*.

**CAL1 positively regulates Cd contents in leaf and xylem sap.** Os02g0629800 (*CAL1*) encodes a putative defensin precursor, consisting of a cysteine-rich domain and a secretion signal peptide (Fig. 1h). Expression analysis showed that *CAL1* was preferentially detected in roots and leaf sheath of rice seedlings (Fig. 1i). When under normal condition, the expression levels were similar between the near isogenic line (NIL) of *CAL1*, NIL (TN1), and its isogenic control, NIL(CJ06). In contrast, Cd exposure significantly induced *CAL1* expression in NIL(TN1) roots (Fig. 1i). Further analysis showed that *CAL1* expressed in various tissues except leaf blades (Fig. 1j), and significant higher expression was observed in node I and the adjoining flag leaf sheath when compared between NIL(TN1) and NIL(CJ06) (Fig. 1j). Moreover, higher Cd accumulation was detected in seedling leaf blades (Fig. 2a) and mature plant straws (Supplementary Fig. 5a) of NIL(TN1) when compared with NIL(CJ06). Increased Cd concentration was also observed in xylem sap of NIL(TN1) (Fig. 2b). However, significant difference was not observed for Cd in grains between NIL(TN1) and NIL(CJ06) (Fig. 2c), and neither the accumulation of essential mineral elements in both the straws and the grains (Supplementary Fig. 5b, c). These results suggest that CAL1 might specifically regulate Cd translocation from roots to shoots, but not from shoots to grains, implying that it is feasible to develop rice varieties over-accumulating Cd specifically in straws meanwhile producing safe grains. Moreover, given higher Cd accumulation occurred in both xylem sap (Fig. 2d) and leaves (Supplementary Fig. 4h) of the complementation lines (N47-14, N47-15, and N47-18) than the control CJ06, while not for Cd contents in roots (Supplementary Fig. 6a) or other metals in leaves (Supplementary Fig. 6b) and xylem sap (Supplementary Fig. 6c), these observations further support a Cd-specific role for CAL1 in rice. In the loss-of-function mutant *cal1* plants generated by CRISPR/Cas9 mutagenesis (Fig. 2e), Cd levels decreased in both leaves (Fig. 2f, g) and xylem sap (Fig. 2h, Supplementary Fig. 7a), while significant effect was not observed in root Cd accumulation (Supplementary Fig. 7b, c). These results together suggest that CAL1 promotes Cd loading into xylem vessels and the consequent accumulation in leaves.

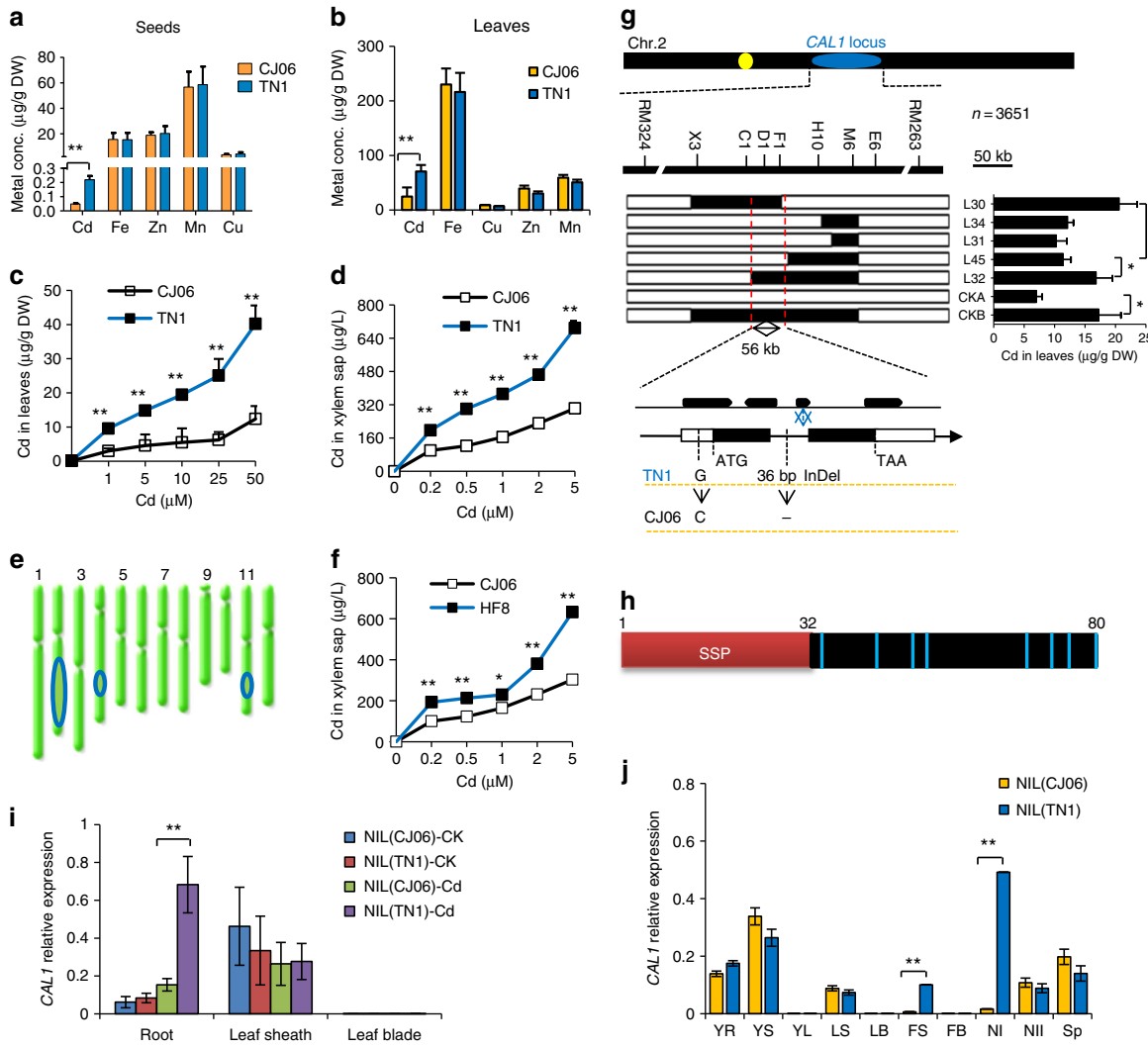

**Fig. 1** Map based cloning and expression of *CAL1*. **a–d** Characterization of metals accumulation and transport in rice cultivars TN1 and CJ06. Metal accumulation in grains from rice grown in contaminated paddy field (**a**), or leaves from seedlings exposed to 10 µM Cd for 7 days (**b**). Cd levels in leaves (**c**) and xylem sap (**d**) from rice seedlings exposed to Cd for 7 days at indicated concentrations. All seedlings were grown in hydroponics to 2-week-old before treatments (**b**, **c**, **d**). **e** QTLs controlling Cd accumulation in rice leaves. Blue ovals indicate the rough genetic intervals that QTLs locate. **f** Cd concentrations in xylem sap from the CSSL line HF8 and the low accumulation cultivar CJ06. **g** Cloning of *CAL1*. Fine mapping (top) and high-resolution mapping (middle) delimited the *CAL1* locus to a 56-kb region. Middle right, foliar Cd contents of recombinant lines L30 and L32 and the control CKB (homozygous for TN1 in the target region) were essentially higher than those of L34, L31, and L45 and the control CKA (homozygous for CJ06 in the target region) under 10 µM Cd for 7 days. Low, gene structure and sequence variation of *CAL1* between TN1 and CJ06. **h** Structural model of the CAL1 precursor. Red box represents the putative secretion signal peptide domain (SSP); black box represents mature protein. Light blue bars indicate cysteine residues, and numbers indicate the sequence number of amino acids. SSP was predicted by web tool signalP 4.1 server. **i** *CAL1* expression in roots, leaf sheath and leaves blade of 2-week-old seedlings treated with Cd at 0 µM (CK) or 10 µM (Cd) for 7days. **j** Expression pattern of *CAL1* in various rice tissues. YR young root, YS young leaf sheath, YL young leaf blade, LS leaf sheath, LB leaf blade, FS flag sheath, FB flag blade, NI node I, NII node II, SP spikelet. Data are mean ± SD, $n = 3$ in (**a–d**, **f**, **i**, **j**), or $n = 5$ in (**g**), and normalized to *Actin1* (**i**) or *HistoneH3* (**j**), respectively. Significant differences were determined by Student's *t*-test (*$P < 0.05$, **$P < 0.01$)

**Cellular and subcellular localization of CAL1**. To answer how CAL1 mediates Cd translocation from roots to shoots, we first determined *CAL1* expression pattern by histochemical analysis of the *CAL1*^TN1 promoter-driven GUS expression. Consistent with the quantitative RT-PCR results, GUS activity was detected mainly in roots, coleoptiles, flag leaf sheath and nodes (Supplementary Fig. 8). Cross-section analysis showed that GUS expression was located preferentially to xylem parenchyma cells in vasculature of root and flag leaf sheath (Fig. 3a, b), as well as in root exodermises (Fig. 3a). The expression pattern of *CAL1* was further confirmed using the construct *CAL1-mRFP* driven by the native *CAL1*^TN1 promoter (Fig. 3c, d), supporting a role of CAL1

to facilitate Cd loading into xylem vessels. Considering the defensin-like CAL1 has a secretion signal peptide at the N-terminus, we then transformed onion epidermal cells with two different constructs *CAL1-mRFP* and *mRFP-CAL1* driven by the 35S promoter. Subcellular localization assay showed that in contrast to the control (Fig. 3e), fluorescence signal was preferentially detected in cell walls when transformed with *CAL1-mRFP* (Fig. 3f). Fluorescence was observed in nucleus and cytoplasm of cells harboring *mRFP-CAL1* (Fig. 3g), indicating that mRFP at the N-terminus of CAL1 might interfere with its secretion signal peptide thus affecting the protein secretion from cytosol to extracellular spaces. Using the construct 35S::*CAL1-*

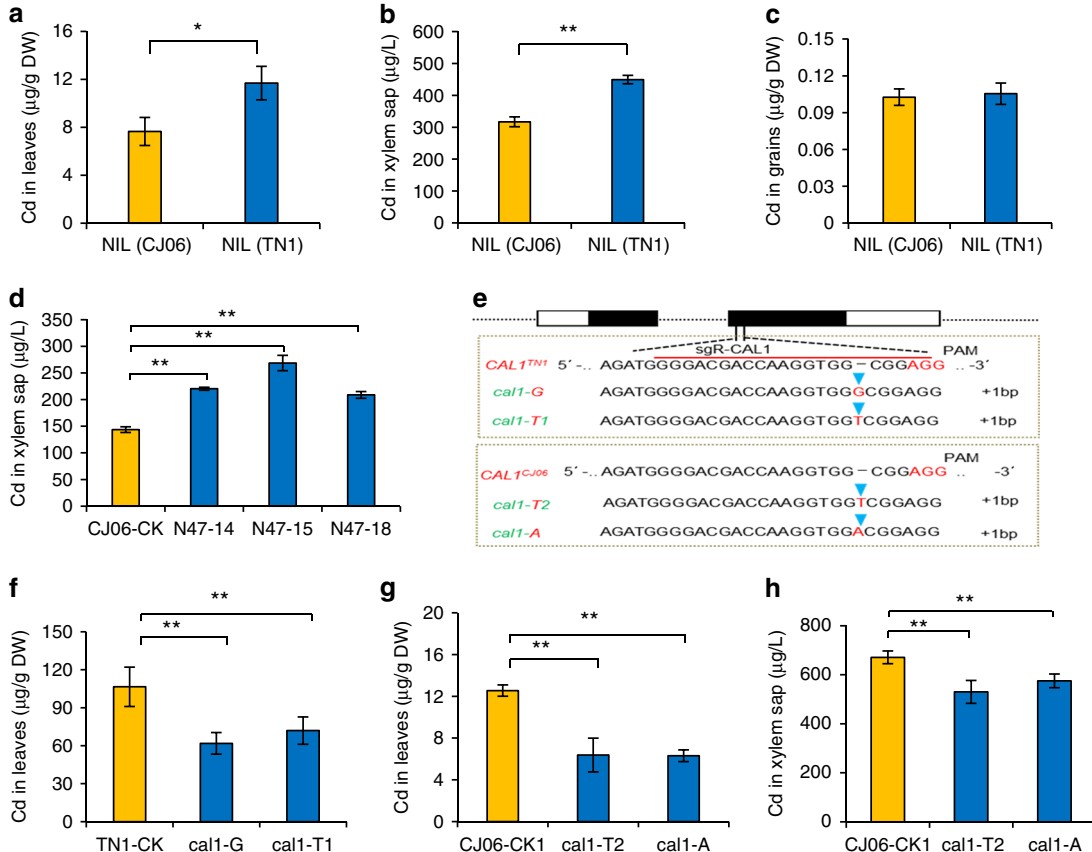

**Fig. 2** CAL1 positively regulates Cd contents in rice leaf and xylem sap. Leaves and xylem sap were collected from 2 weeks old seedlings exposed to 10 μM Cd for 7days, and grains was sampled from rice plants grown in Cd-contaminated paddy soil. **a–c** Cd levels in leaves (**a**), xylem sap (**b**) and grains (**c**) from NIL(TN1) and NIL(CJ06). **d, e** determination of Cd contents in xylem sap (**d**) of the *CAL1*-complementation lines N47-14, N47-15, and N47-18, which simulate overexpression lines due to increased expression of *CAL1* (Supplementary Fig. 3d) in CJ06 background. CJ06-CK represent transgene-negative controls. **e** Identification of *cal1* mutants generated by CRISPR/Cas9. *cal1-G* and *cal1-T1* represent mutants with single G or T insertion in *CAL1* coding region, respectively, with TN1 background. *cal1-T2* and *cal1-A* represent mutants with single T or A insertion in CAL1 coding region, respectively, with CJ06 background. **f–h** Cd contents in leaves (**f, g**) and xylem sap (**h**) of *cal1* mutants. TN1-CK and CJ06-CK1 represent transgene-negative controls. Data are mean ± SD, n = 3 in (**a–c**), or 4 in (**d** and **f–h**). Significant differences were determined by Student's *t*-test (*$P < 0.05$, **$P < 0.01$)

*mRFP*, subcellular localization of CAL1 was also performed *in planta*, and a similar localization in cell walls was observed in rice root cells undergoing plasmolysis (Fig. 3h), noting the CAL1-mRFP fusion proteins are functional in plants (Supplementary Fig. 9a–d). Given defensin proteins are cysteine-rich and contain eight thiols[10, 11], these data suggest that CAL1 might function to chelate Cd, thus driving Cd efflux from xylem parenchyma cells to extracellular spaces and the consequent loading into xylem sap.

**Functional characterization of CAL1**. To test this hypothesis, in vitro metal binding assays were performed and two pH values were used to simulate the acidity in the cytosol (7.5) or in the xylem sap (5.5). A CAL1/Cd stoichiometry of ~1.2 was consistently detected under both pH values for the SSP (secretion signal peptide)-deleted ΔSPCAL1, which simulated mature CAL1 to be secreted to apoplastic compartments. While for the full length CAL1, the CAL1/Cd stoichiometry was 0.98 at pH 7.5 or 0.63 at pH 5.5 (Fig. 4a), which are still significantly different from the control tagging protein TF (Trigger Factor). These data suggest that both the full length and the mature CAL1 proteins can bind Cd, but the mature one is more efficient; pH change does not affect CAL1's binding ability during its secretion from the cytosol to the apoplast. Cd binding to CAL1 was further confirmed by isothermal titration calorimetry (ITC) assay, which

showed a dissociation constant/$K_D$ of 53.7 μM (Supplementary Fig. 10). Using site-directed mutagenesis, we revealed that only 3 cysteine residues at Cys55, Cys65, and Cys75 significantly decreased the Cd binding ability of CAL1 (Supplementary Fig. 11 and Supplementary Table 4). In addition, CAL1 did not bind significant amounts of the competing ions Ca, Mn, or Zn (Supplementary Fig. 12). Taken together, these data suggest that Cd is possibly coordinated to three thiol groups in CAL1 to form a stable Cd: 3(SH−) complex, similar to that observed in phytochelatins[21].

Ectopic expression of ΔSPCAL1 enhanced Cd tolerance and accumulation in both E. coli and yeast (Supplementary Fig. 13), consistent with the model that CAL1 chelates Cd and efflux of the complex is dependent on the secretion signal peptide. Further analysis showed that Cd levels in CJ06 protoplasts were higher than those of the complementation lines N47-14 and N47-15 (Fig. 4b), in which higher *CAL1* expression was observed compared to CJ06 (Supplementary Fig. 4d). When incubated in W5 buffer for 6 h, Cd levels decreased faster in N47s' protoplasts than in CJ06's (Fig. 4b), while in the function-disrupted *cal1* mutant plants (*cal1*-T2 and *cal1*-A) (Fig. 4c), the decline rate was slower than in the control CJ06. Cd extrusion was significantly enhanced in yeast expressing CAL1 compared to those expressing the non-secreted form or the empty vector (Supplementary Fig. 14). These data consistently support our hypothesis that

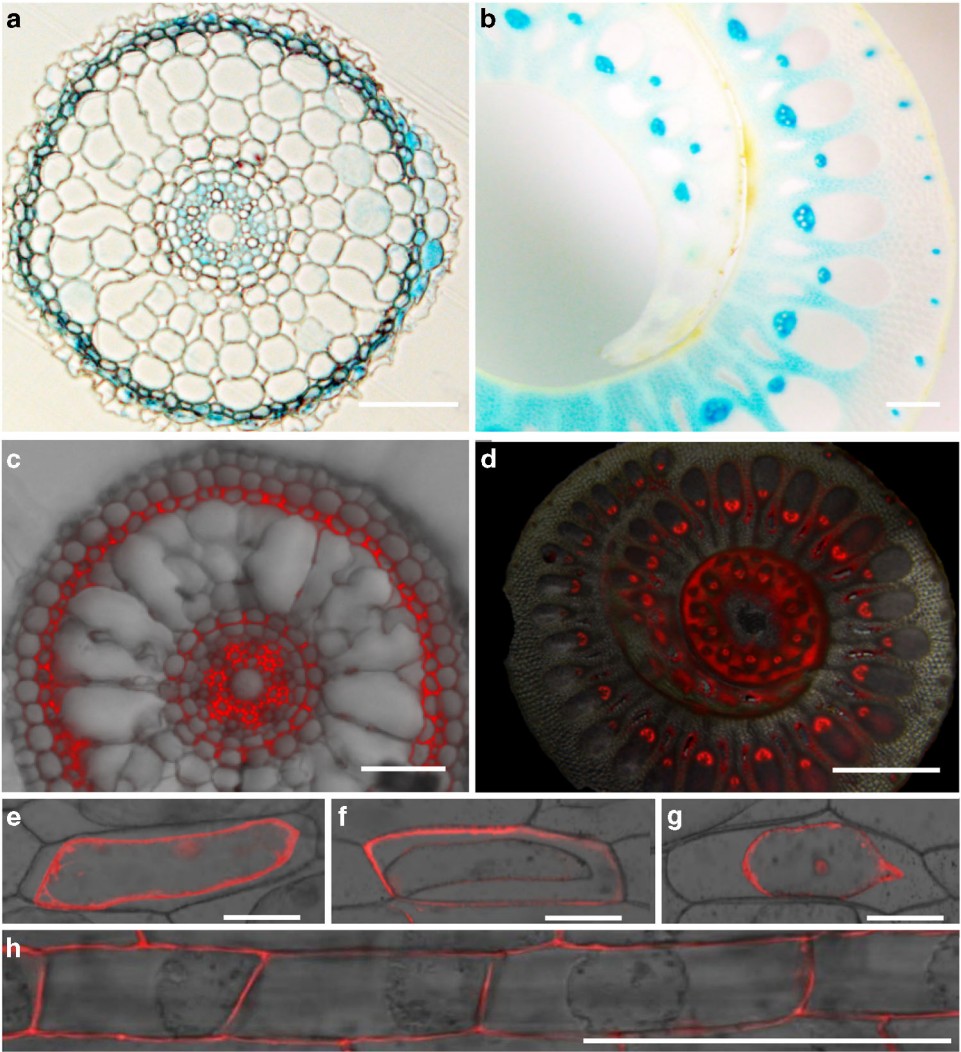

**Fig. 3** Histochemical and subcellular localization assay. **a**, **b** GUS expression driven by the *CAL1*^TN1 promoter. (**a**) Cross section of 1 week old seedling roots. **b** Cross section of flag leaf sheath. **c**, **d** Fluorescence images of cross-sectioned seedling roots (**c**) and flag leaf sheath (**d**) from plants expressing the construct pro*CAL1*^TN1::*CAL1-mRFP*. **e**–**g** Subcellular localization of CAL1. Onion epidermal cells transiently transformed with *mRFP* (**e**), *CAL1-mRFP* fusion (**f**), or *mRFP-CAL1* (**g**) were incubated in 40% sucrose to induce plasmolysis and then imaged by confocal microscopy. **h** Subcellular localization of CAL1 in rice plants harboring the construct 35S::*CAL1-mRFP*. Root epidermal cells were incubated in 40% sucrose to induce plasmolysis and then imaged by confocal microscopy. Bars = 1 mm in (**b**, **g**), 100 μm in (**a**, **c**–**e**, **f**, **h**)

CAL1 functions most likely to chelate Cd and hence facilitating Cd efflux from protoplasts. More directly, mass spectrometry analysis detected fragments of CAL1 protein in xylem sap (Supplementary Fig. 15). Western blotting further detected the CAL1 fusion protein in xylem sap of P78 plants harboring the pro*CAL1*^TN1::*CAL1-mRFP* construct (Fig. 4d). To rule out the possibility that xylem sap collected from cut stem could be contaminated by the symplastic content of the cut cells, we further collected guttation sap from the leaves of intact plants, which represents uncontaminated apoplastic fluid[22, 23]. The results showed that CAL1 was also detected in guttation fluid of intact rice leaves (Supplementary Fig. 16). These data indicate that the mature CAL1 was extruded from the cytosol and subsequently underwent long-distance transport in xylem vessels, likely driven by transpiration stream. Interestingly, an additional band (~34 KD) was detected in leaf blade and sheath tissues compared to xylem sap (Fig. 4e), indicating that CAL1 presented also in full length protein form in these tissues. Given the full length fusion protein (~34 KD) was much less than the mature one (~30 KD) in leaf blades while similar levels were detected in

leaf sheath (Fig. 4e), and the *CAL1* transcript expression was almost undetectable in leaf blade but much higher in leaf sheath (Fig. 1j), these observations together suggest that CAL1 protein in leaf blade is preferentially in mature form and translocated from outside, consistent with our model that mature CAL1 is extruded from xylem parenchyma cells into xylem sap and subjected to long-distance transport, hence increasing Cd accumulation in aerial tissues. This model was further supported by ectopic expression of CAL1 in *Arabidopsis*, where significantly increased Cd contents were observed in both shoots and xylem sap (Supplementary Fig. 17). It is worth noting that even when no Cd was added, the mature CAL1 protein was still detected in xylem sap (Fig. 4d, left two lanes), indicating that CAL1 might have other functions yet to be identified.

Given *CAL1*^TN1 and *CAL1*^CJ06 showed bunch of variations in the promoter sequences thus it is hard to find out which mutation (s) contribute to the enhanced *CAL1* expression in TN1, we then sequenced 107 rice germplasms in the mini Core Collection, trying to find other *CAL1* haplotypes with enhanced expression levels and mutations in their promoter regions, but the elevated

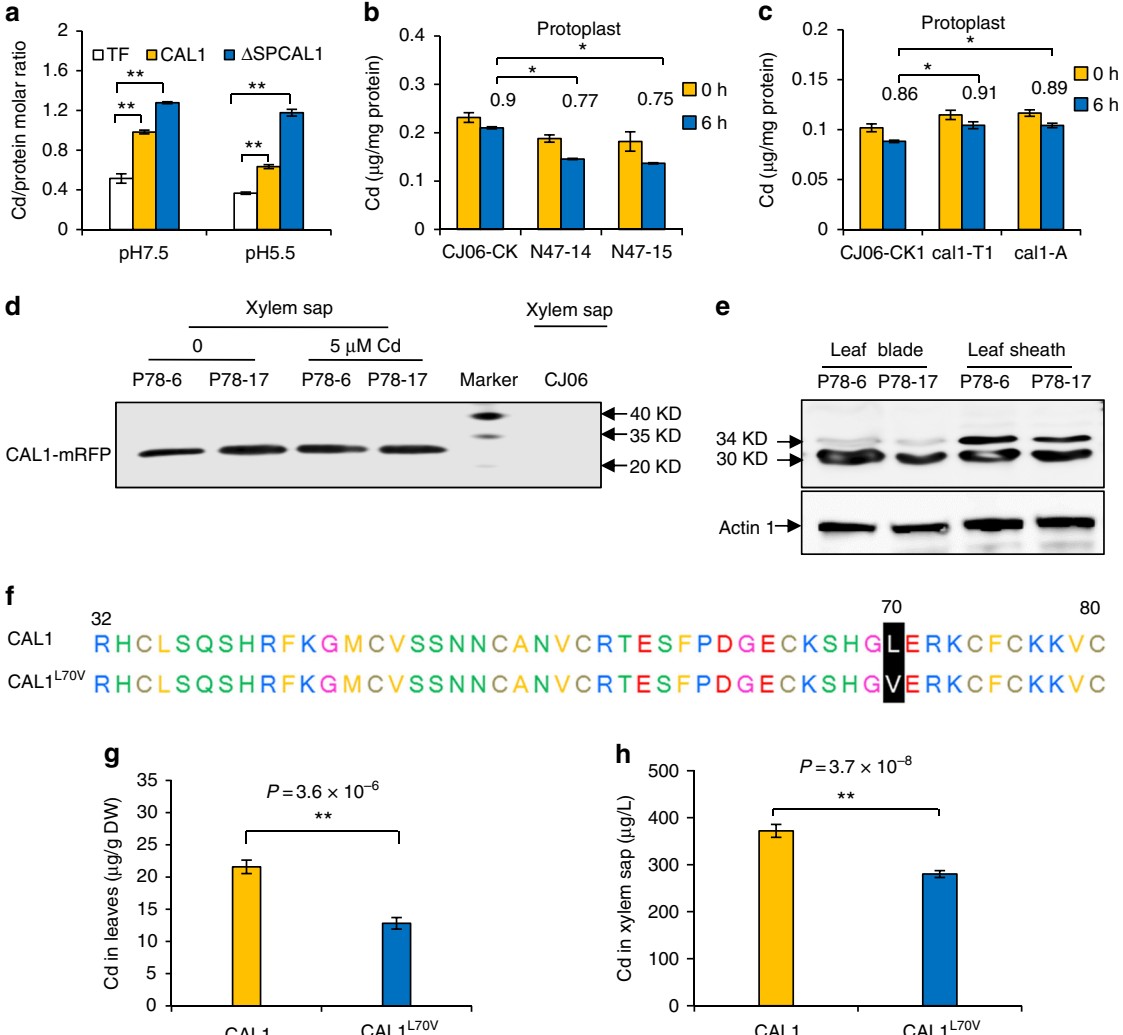

**Fig. 4** Functional characterization of CAL1. **a** In vitro Cd binding assay. Full length CAL1 and the secretion signal peptide (SP)-deleted CAL1 (ΔSPCAL1) were used. Different form of CAL1 recombinant proteins were extracted from BL-21, and then incubated with 100 μM Cd for 1 h, pH = 7.5 or 5.5. TF represents the *E. coli* trigger factor protein that fused to the N-terminus of target proteins. **b**, **c** Cd contents in protoplasts. Rice seedlings of the complementation lines (**b**) or functional disrupted mutants (**c**) were grown for 14 days in hydroponics supplemented with 5 μM Cd, and leaf sheath was sampled to isolate protoplasts, which were then cultivated in W5 buffer for indicated time before determination of Cd contents. The numbers above each bar represent the 6/0 h Cd ratios in protoplasts, and significant difference between them was determined by Student's *t*-test (*$P < 0.05$). Data are mean ± SD, $n = 3$. **d**, **e** Detection of CAL1 by using an anti-RFP antibody in transgenic plants P78 harboring the construct proCAL1$^{TN1}$::CAL1-mRFP. Rice seedlings were grown for 21 days in hydroponics supplemented with 0 or 5 μM Cd. Proteins extracted from xylem sap (**d**) or leaf and stem tissues (**e**) were subjected to Western gel blotting assay. The wild-type CJ06 was used as a negative control. Actin was used as the loading control. **f** Two types of CAL1 proteins derived from natural variations in different rice varieties. **g**, **h** Cd contents in leaf (**g**) or xylem sap (**h**) of rice varieties harboring either CAL1 ($n = 100$) or CAL1L70V ($n = 7$). Two-weeks-old seedlings grown in hydroponics were exposed to 10 μM Cd for 7 days, and then subjected to sampling and determination of Cd contents. Data are mean ± SE. Significant differences were determined by Student's *t*-test (*$P < 0.05$, **$P < 0.01$)

expression and mutations did not always correlate to elevated Cd accumulation, which is most probably due to the highly different genetic background. Interestingly, one conserved variation in coding sequence was detected in 7 out of the 107 germplasms, which led to the transition of Leu to Val in CAL1 at amino acid site 70 (CAL1$^{L70V}$, Fig. 4f). Coincidently, Cd levels in leaves (Fig. 4g) or xylem sap (Fig. 4h) were significantly lower in those 7 CAL1$^{L70V}$-type varieties than in the rest 100 varieties, implying that the transition of L70V in CAL1 resulted in less long-distance transport of Cd to aerial parts. To explore the underlying mechanisms, we first determined the subcellular localization of CAL1$^{L70V}$, and the data said that CAL1$^{L70V}$ could still be secreted into extracellular compartments (Supplementary Fig. 18a). Further in vitro binding assay showed that CAL1$^{L70V}$'s binding

capacity for Cd was probably disrupted (Supplementary Fig. 18b). These results suggest that impaired Cd binding might contribute to Cd under-accumulation in leaves of those varieties harboring CAL1$^{L70V}$, and CAL1$^{L70V}$ might serve as a marker for marker-assisted breeding of Cd under-accumulation varieties.

## Discussion

In this study, we identified to our knowledge the first QTL (*CAL1*) that specifically mediates Cd efflux from cytosol into extracellular spaces via chelation. Previous studies showed that transporter genes contribute essentially to vacuolar sequestration[24–32] or direct xylem loading of Cd[33–39], thus mediating long-distance transport of Cd and other transition metals between

different tissues. However, our data showed that the defensin-like protein CAL1 might specifically bind Cd in cytosol, and such a binding drives Cd secretion from xylem parenchyma cells into the xylem vessels, hence lowering Cd levels in cytosol meanwhile promoting Cd translocation from roots to shoots, though it remains to be determined if transporters or vesicular trafficking pathways are responsible for translocating the CAL1:Cd complex across the plasma membrane. Given ~20% decrease of Cd was observed in xylem sap of the *cal1* mutant compared with the wild type control, we postulated that CAL1 might contribute to ~20% of the Cd translocation rate, which is reasonable as this QTL was estimated to contribute ~13% of the overall variance. Moreover, it appeared that the CAL1:Cd complex could not be taken back into symplastic compartments (Fig. 4b), which actually prevents Cd loading into phloem, the major pathway for Cd accumulation in rice grains[40–43]. This hypothesis might provide an explanation of why CAL1 specifically affected Cd accumulation in leaf blades and rice straws, but not in grains. More interestingly, the CAL1-mediated Cd accumulation might represent a novel and ideal mechanism applicable to the development of remediation-rice: over-accumulating Cd in rice straws to remediate polluted soils, meanwhile producing safe, and nutritional rice grains without toxic levels of Cd. This kind of remediation-rice would be highly demanded in countries like China, as most of the moderately polluted paddy soils there have to be used to produce foods due to the huge population and relative shortage of land.

## Methods

**Plant materials and growth conditions**. Doubled haploid (DH) population and related genetic map were constructed in Qian lab using parental varieties Tainan1 (TN1) and Chunjiang06 (CJ06)[20]. CSSLs were constructed with TN1, the variety showing Cd over-accumulation, as the donor parent and the Cd under-accumulation variety CJ06 as the recurrent parent. HF8 is the CSS line containing the *CAL1* segment from TN1 (*CAL1*[TN1]) and showed higher Cd accumulation in leaves. Meanwhile, a NIL of *CAL1*, NIL(TN1), and its isogenic control NIL (CJ06) were generated by repetitive backcross (BC$_4$F$_4$) and marker-assisted selection. NIL (TN1) harbored an approximately 200-kb *CAL1*-containing chromosome segment from TN1 in the genetic background of CJ06. *Arabidopsis* ecotype Columbia-0 (Col-0) was used for transformation.

Rice plants were grown in paddy fields polluted by heavy metals to harvest seeds. Alternatively, seeds were pretreated and uniformly germinated seeds were planted in 96-well plates with the bottoms removed[44]. Rice seedlings were grown hydroponically in Yoshida solution (pH 5.8) at 28 °C, 60% relative humidity with a 13-h light/11-h dark photoperiod. *Arabidopsis thaliana* plants were grown in quarter-strength sterile hydroponic solution at 22 °C with 16-h-light/8-h-dark cycles[45].

**QTL analysis and fine mapping of *CAL1***. Rough QTL mapping was performed by map marker 3.0/QTL soft to create genetic linkage map and to calculate QTL loci[46], using 119 DH lines and 177 sectioned seedling roots molecular markers. To map the *CAL1* locus precisely, fine mapping was performed using the BC$_3$F$_2$ population derived from HF8 and CJ06, which localized *CAL1* to the interval between markers RM324 and RM263. Sequence data for CAL1 can be found in the GenBank/EMBL databases under the accession number MF580139. High-resolution mapping of *CAL1* was performed using 3651 BC$_3$F$_3$ plants and 14 markers newly developed. Homozygous recombinant plants (BC$_3$F$_4$) were identified and used to determine Cd accumulation in leaves of their progenies derived from self-pollination. The *CAL1* locus was ultimately localized to a 56-kb region between markers C1 and F1. Genomic DNA of candidate gene was amplified and sequenced to identify genetic variations. Primers for mapping and amplification of DNA fragments were listed in Supplementary Table 5.

**Plant sampling and elemental determination**. At 2 weeks of age, the hydroponically grown rice seedlings were exposed to Cd treatments as indicated before the second leaves were sampled. Seedling shoots were cut off by a sharp razor to collect xylem sap for 2 h, and the first drop of xylem sap was discarded to avoid contamination from damaged cells. Xylem sap collected from 16 plants was pooled into one replicate, and a total of three replicates were used for each line. Four weeks old *Arabidopsis* plants were exposed to 10 μM Cd for 3 days before tissue sampling and xylem sap collection. Briefly, all rosette leaves were removed, the inflorescence stems were cut using a sharp razor, and xylem sap was collected for 2 h[47]. Metal contents were determined by inductively coupled plasma mass spectrometry (ICP-MS) as follow:heavy-metal-treated and control plant tissues were digested in 70%

nitric acid for 3–5 days at room temperature, and then samples were boiled until completely digested. Samples were diluted with Millipore filtered deionized water and briefly centrifuged[45].

**DNA constructs and transformation into plants**. A 1974-bp *CAL1* promoter fragment in TN1 genomic sequence was amplified by PCR using primers Pro-CAL1F and ProCAL1R (Supplementary Table 5), the resulted pro*CAL1*[TN1] promoter fragment was subcloned into the binary vector pCAMBIA1300[48]. The complementation construct was generated by PCR amplification of a 2635-bp fragment (containing 1974-bp promoter sequence before ATG, the 343-bp full length ORF and the 318-bp 3′-UTR of *CAL1* from TN1 genomic DNA) using primers HB-CAL1F and HB-CAL1R (Supplementary Table 5) and subcloning it into the binary vector pCAMBIA1300. To determine the subcellular localization of CAL1 in rice, the *CAL1-mRFP* fragment was recovered from the construct 35S::*CAL1-mRFP*/PA7 and inserted into pCAMBIA1300 to obtain the 35S::*CAL1-mRFP*/pCAMBIA1300 construct, which was further reconstructed by replacing 35S promoter with the native promoter pro*CAL1*[TN1], resulting a new construct pro-*CAL1*[TN1]::*CAL1-mRFP*/pCAMBIA1300. To generate *cal1* mutants, the *CAL1*-specific guide RNA expression sequence was introduced into the CRISPR-Cas9 construct[49] using primers sgR-CAL1F and sgR-CAL1R (Supplementary Table 5), and the resulted 1300-bp fragment of Cas9 and *CAL1*-specific guide RNA expression cassettes were recovered by *Hind*III/*Eco*RI restriction digestion and subcloned into pCAMBIA1300[49]. All the resulted constructs were introduced into *Agrobacterium tumefaciens* strain EHA105 or GV3101, and transformed into CJ06 or TN1 as described[50] by Wuhan Bo Yuan Biotech. Co. (Ltd). Alternatively, the 35S::*CAL1-mRFP*/pCAMBIA1300 construct was transformed into *Arabidopsis* using the floral dip method[51]. Transgenic plants were screened using primers HYGF and HYGR (Supplementary Table 5) and confirmed by quantitative PCR.

**Expression and histochemical analysis**. Rice plants were grown in hydroponics to 2 weeks old before treatments as indicated, and then sampled as indicated. Alternatively, rice plants grown in paddy fields were subjected to tissue sampling at heading stage. Total RNA was extracted from those collected tissues using TRIZOL reagent as suggested by manufacturer (Invitrogen). First-strand cDNA synthesis and quantitative RT-PCR were performed. Primers used in these assays are listed in Supplemental Table 5, and the expression levels were normalized to those of the *HistonH3* or *Actin1* as indicated. To confirm the histological expression pattern of GUS driven by the pro*CAL1*[TN1] promoter, GUS histochemical staining was performed for 8 h with 2 mM X-Gluc (5-bromo-4-chloro-3-indolyl-b-D-glucuronide), 5 mM K$_4$Fe(CN)$_6$, 5 mM K$_3$Fe(CN)$_6$, 0.2% Triton X-100, and 50 mM NaPO$_4$, pH 7.2 (68.4 parts of Na$_2$HPO$_4$ with 31.6 parts of NaH$_2$PO$_4$). The reaction was stopped by 70% ethanol and then photographed[47].

**Subcellular localization and protoplast isolation**. The fragment of *mRFP* was amplified by PCR using primers RFPF and RFPR (Supplementary Table 5) and subcloned to generate the construct 35S:: *mRFP*/PA7. The coding sequence of *CAL1* was further amplified using PCR primers 5′CAL1-RFPF and 5′CAL1-RFPR or 3′CAL1-RFPF and 3′CAL1-RFPR (Supplementary Table 5), and the resulted fragments were fused in-frame to the 5′- or 3′- terminus of *mRFP*, respectively, to generate the constructs 35S::*CAL1- mRFP*/PA7 and 35S:: *mRFP-CAL1*/PA7. All constructs were then transiently expressed in onion epidermal cells using a particle gun–mediated system (PDS-1000/He; Bio-Rad). The bombarded cells were held in the dark at 28 °C for 16 h followed by mRFP imaging using confocal microscopy (Carl Zeiss; LSM510Meta). Alternatively, the 35S::*CAL1-mRFP*/pCAMBIA1300 construct was transformed into CJ06, and indicated tissues of the transgenic lines were subjected to mRFP imaging. For protoplast isolation, rice plants were grown to 2 weeks old in hydroponics supplemented with 5 μM Cd. Leaf sheaths from about 30 seedlings were cut into approximately 0.5 mm strips with propulsive force using sharp razors to isolate protoplasts[52]. Metal concentration in protoplasts were determined by ICP-MS and normalized to corresponding total proteins.

**Protein purification and metal binding assay**. Fragments of *CAL1* (full length coding sequence), Δ*SPCAL1* (a truncated *CAL1* representing mature CAL1 protein at amino acid sites 32–80, without the secretion signal peptide at amino acid sites between 1–31) and Δ*SPCAL1*[L70V] (truncated *CAL1* with a natural variation that led to the transition L70V) were amplified by PCR using primers TF-CAL1F and TF-CAL1R or Δ*SPCAL1F* and Δ*SPCAL1R* (Supplementary Table 5) and cloned into the vector pCold-TF, respectively. Site-specific mutagenesis of Δ*SPCAL1* was performed using fast mutagenesis system (TRAN, #FM111). Primers for Site-specific mutagenesis were listed in Supplementary Table 5. Δ*SPCAL1* was also cloned into pGEX-2TK for metal sensitivity assay in *Escherichia coli*. The resulting constructs were further transformed into *E. coli* strain *BL21 (DE3)*. Recombinant proteins were induced by 0.2 mM isopropyl b-D-1-thiogalactopyranoside at 16 °C for 19 h, and the cells were harvested and lysed using a French press. Cell debris was removed by centrifugation. The supernatant was purified using Ni-NTA agarose (Qiagen), and further eluted and concentrated[53].

Metal binding was determined using two methods. Briefly, 100 μg purified protein was incubated with 100 μM Cd in 100 μl buffer (20 mM Tris-HCl, 100 mM NaCl, pH7.5 or 5.5) for 1 h at 20 °C, then unbound Cd was removed by Zeba Spin

Desalting Columns (Thermo Scientific), which through centrifuge collected metal protein complex for ICP-MS. Alternatively, isothermal titration calorimetry (ITC) experiments were performed with a MicroCal ITC200 system (Malvern) at 20 °C in a buffer containing 20 mM Tris-HCl, pH 7.5, 100 mM NaCl. 1 mM metal and 100 µM protein were used for ITC assay[53] with minor modification.

**Western blotting and ESI-MS analysis.** To test if CAL1 is subjected to long-distance transport, ProCAL1$^{TN1}$::CAL1-mRFP was transformed into CJ06, and the transgenic plants were grown in hydroponic solution (spiked with/out 5 µM Cd) to 21 days old before tissue and xylem sap sampling. Xylem sap was also sampled from the complementation line N47 grown under the same condition. Total proteins were extracted from tissue samples using buffer E (125 mM Tris-HCl, pH 8.0, 1% [w/v] SDS, 10% [v/v] glycerol, and 50 mM NaS$_2$O$_5$). Thirty microgram total proteins from each tissue sample or 30 µL xylem sap were separated on 10% SDS-PAGE gel and subjected to Western gel blotting according to the manu-facturer's standard protocol. Mouse anti-mRFP (Kang Wei, CW0254M; at 1:2000 dilution) or anti-Actin (Kang Wei, CW0264M; at 1:2000 dilution) were used as primary antibodies. Horseradish peroxidase labeled anti-mouse antibody (Ding Guo, SH00-12; at 1:5000 dilution) was used as a secondary antibody. Membranes were visualized using an Immobilon Western Chemiluminescent HRP Substrate Kit (MILLIPORE) and photographed with Image Quant LAS 400 mini. Full images of western blot were shown in Supplementary Fig. 19. ESI-MS detection of proteins in xylem sap was done by Shanghai Applied Protein Technology, Ltd.

**Yeast complementation assay.** The truncated ΔSPCAL1 and full length CAL1 were cloned into pYES2. The resulted constructs were transformed into the yeast (Saccharomyces cerevisiae) mutant strain Δyap1[54] (MATα ura3 lys2 ade2 trp1 leu2 yap1::leu2)[55]. For metal sensitivity assays, yeast cells were grown for 16 h in liquid synthetic defined (SD) medium with appropriate supplements, yeast (Δyap1) cells were harvested and diluted as indicated and then spotted onto SD medium-containing plates supplemented with 2% galactose and 40 µM Cd[27]. The plates were incubated at 30 °C for 5–7 days before photographed.

**Data availability.** The authors declare that all data supporting the findings of this study are available within the article and its Supplementary Information files or are available on request to the corresponding authors.

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

## Acknowledgements

We thank Drs. J. K. Zhu (Shanghai Center for Plant Stress Biology, Chinese Academy of Sciences) for providing CRISPR-Cas9 construct, W. H. Tang and D. Y. Chao (Shanghai Institute of Plant Physiology and Ecology, Chinese Academy of Sciences) for providing the mRFP fragment and the pCold-TF vector, X. S. Gao (Core Facility, Shanghai Institute of Plant Physiology and Ecology, Chinese Academy of Sciences) for helping with confocal microscopy, P. Zhang (Shanghai Institute of Plant Physiology and Ecology, Chinese Academy of Sciences) for helping with ITC analysis. This work was supported by the National Natural Science Foundation of China (31325003 and 31421093), the National Key R&D Program of China (2016YFD0100700) and the XDPB0402 of Chinese Academy of Sciences.

## Author contributions

J.-M.G. conceived and supervised the project. J.-M.G., J.-S.L., and J.H. designed the experiments. J.-S.L., and J.H. performed most of the experiments. D.-L.Z., G.-B.Z., J.-S.P., H.-L.M., Y.G., H.-Y.Y., and Q.Q. performed some of the experiments. J.-S.L., J.H., Y.-L.F., B.H., H.-X.L., and J.-M.G. analyzed data. J.-S.L., and J.-M.G. wrote the manuscript.

## Additional information

**Competing interests:** The authors declare no competing financial interests.

