## [Peer Review File · Nature Communications]

Reviewers' comments:

Reviewer #1 (Remarks to the Author):

Manuscript reference: NCOMM-122849_0

Title: Defensin-driven cadmium secretion and allocation in rice

Authors: Jin-Song Luo, Jing Huang, Da-Li Zeng, Guo-Bin Zhang, Jia-Shi Peng, Hai-Ling Ma, Yuan Guan, Hong-Ying Yi, Yan-Lei Fu, Bin Han, Hong-Xuan Lin, Qian Qian, Ji-Ming Gong

From an analysis of a collection of rice accessions, the authors identified one accession (TN1) that accumulated high levels of Cd both in its leaves and grains. They crossed this accession with another one (CJ06) that showed lower cadmium contents. A QTL analysis delivered 3 loci controlling Cd accumulation in the shoot. The authors selected one of the loci and made the positional cloning of this locus, which showed that a defensin-like protein (CAL1) was responsible for the variation in Cd accumulation. Interestingly, the TN1 allele is responsible for higher accumulation of Cd in the leaves and the xylem sap, but not in the grains, opening the possibility to breed rice varieties that can extract Cd from the soil while preserving the quality (and thus the commercialization) of the grain. CAL1 transcripts are shown to be induced by cadmium. CAL1 is shown to be mainly expressed in roots, coleoptiles, flag leaf sheath and nodes. In roots, the expression occurs in exodermis and xylem parenchyma cells. The protein is present in cell walls.

CAL1 is proposed to specifically bind cadmium (and neither of Fe, Zn, Mn, Cu, Ca or Mg). Protoplasts of rice transgenic plants expressing the TN1 allele of CAL1 are shown to accumulate less cadmium over time than protoplasts of controls or of CAL1 mutants.

From these results (and a couple of additional ones), the authors claim that they identified a novel mechanism involved in the secretion of Cd from the cell to the extracellular space including the xylem, and that this contributes to the transfer of cadmium from the roots to the shoots and thus to a comparatively higher accumulation of Cd in the leaves. This novel mechanism would be the binding of cadmium to a defensin-like protein that is excreted from the cell.

The data reported here are for sure novel. They will be of large interest because, besides interesting the research community working on metal transport and accumulation in plants, they point a role of defensin-like proteins in a process in which these proteins were up to now not expected to play a role (at least to my knowledge). As such, these results should contribute to bring new perspectives and make thinking evolve in different fields of research. In addition, these results show that it could be possible to both increase the accumulation of cadmium in a plant while 'preserving' the grain, and thus this work should lead to interesting practical outputs.

In a large part because the authors made efforts to bring different pieces of evidence to support their conclusions the work is as a whole convincing. Although, specific points should be improved. The genetic part and the demonstration that CAL1 is responsible for the phenotype of interest both appear clear. The localization of the expression of the gene seems clear, although the authors only exploit the fact that the gene is expressed in the xylem parenchyma in roots and not the fact that it is also expressed in other tissues. Concerning the analysis of the protein, I have a major concern about the western analyses (Figure 4). To my knowledge, the mRFP alone is already heavier than 30 kDa. When it is fused to the defensin-like protein, it should be at least 5 kDa heavier. This is not compatible with the results presented in Figure 4. I thus wonder whether the authors indeed analyzed the defensin-like protein or a cleaved mRFP (this occurred at least once in my lab) on their western blots. Thus this may question the actual subcellular location of the protein since it is detected only through the analysis of the mRFP.

I am not familiar with protein-metal binding assays, but I feel that the authors should improve the quality of the data or of the presentation. The protocol of the in vitro binding assay should be given

entirely (in supplementary material if it is too long). In my opinion, the experiment cannot be reproduced, as it stands. The quantity of Cd that is bound to the protein should be expressed in a mol/mol ratio instead of in a mass/mass ratio. Indeed, if the mol/mol ratio is far below 1 or higher than let's say 2, this should question the significance of the results. I am not sure that the 'tagging protein' is a good negative control. I think that any cystein-rich protein would bind cadmium as CAL1 does, wouldn't it ? The protoplasts assays should be done after equivalent durations of incubation (see details below). The positive point that I note is the metal specificity of the 'binding' but this may be a bit weak.

I am also not really convinced by the fact that the TN1 and CJ06 rice accessions display the same ability to uptake Cd (Supplementary Figure 1). Is the assay really measuring the Cd uptake ? Indeed, if TN1 translocates more Cd from the roots to the shoot than CJ06 while having the same concentration of Cd in the roots (Supplementary Figure 5), it should take up more Cd than CJ30. This should be commented in greater details.

The authors found that a defensin-like protein is involved in the transport and accumulation of cadmium. But what is a defensin ? Why do the authors not give a single word about defensins ? Why do the author name their protein 'defensin' although they have not checked that CAL1 indeed has a defensin activity. They have to use the terms 'defensin-like protein' instead of 'defensin'. At first glance, the result may appear surprising. One could have expected transporters or regulators to have been involved. It is surprising that the authors do not comment about the fact that a defensin-like protein would play the role that they identified. I made a rapid bibliography survey and found that plant and animal defensins have actually already been described as being related to metals (metal tolerance, metal binding ...). I think that it is necessary that the authors connect their findings with data already published showing that defensins are involved in 'metal physiology'.

I wonder why the authors screened the rice collection for variants in the coding sequence (which is not a bad idea per se) and not in the promoter (or intron) to catch similar mutations to those found in TN1 and that would be responsible for the Cd-induction of the expression and the high cadmium accumulation. Is TN1 the only rice accession harbouring mutations in the CAL1 promoter ? What is the phenotype of other accessions showing the same variations in their promoter sequences ?

More specific issues:

- l. 47-48: It is necessary that the authors specify which core collection they screened. They referred to a paper from Zhang et al. (2011) but this is not sufficient since (1) several different core collections are presented in that paper and (2) no "China Rice Core Collection" is defined in that paper. The authors thus have to mention whether they screened the 932 accessions constituting the Core Collection, or the 189 accessions constituting the mini Core Collection, or any other number of accessions.

- l. 47-48: It is absolutely necessary that the authors give a Supplementary Table with the Cd, Fe, Zn, Mn and Cu grain and leaf content measurements in all the accessions that they screened.

- l. 51: the authors should mention in the text the relative difference between Cd content in grain and leaves of the TN1 cultivar compared to the CJ06 cultivar. The reader should not have to go to Figure 1 to see the difference, which is actually not quantified in that Figure.

- l. 58: since the paper describing the doubled haploid population is not easily available, I ask the authors to mention the number of doubled haploids that they used for their QTL approach. I also ask them to mention how many markers they used for the QTL mapping.

- l. 143, as well as in the legends of Figure 4 (l. 536), of Supplementary Figure 9 and of Supplementary Figure 13: what is the (control) 'tagging protein' ?

- In Figure 4, I wonder why the authors have used a 6-hour-long incubation time before evaluating the Cd content in protoplasts of the complementation lines, while they used a 12-hour-long incubation time for the protoplasts of the mutant lines. Considering that they observe a difference at the end of

the 6-hour-long incubation time but not at the end of the 12-hour-long incubation time, is it really the different origins of the protoplasts that makes the difference (as the author claim; l. 152-154) or the length of the incubation time ? I consider that the authors should show data obtained at the end of the same length of incubation time.

- l. 158-159 and Legend of Figure 4: it is absolutely necessary that the authors mention in the legend of Figure 4 that they used an anti-RFP antibody to check the presence of the CAL1::mRFP fusion protein in the xylem sap or the leaf blade and sheath.

- In Figure 4, it would be much better to express the Cd content supposedly chelated to the defensin-like protein in mol / mol (instead of in mass / mass).

- l. 162-167: the English should be improved. I am not sure to have properly understood the point raised by the authors.

- Legend of Supplementary Figure 1: Is it the whole plants that are maintained at 4°C, or the root systems only ?

- Legend of Supplementary Figure 3: TN1 is NOT a Cd hyper-accumulating accession. The concept of hyper-accumulation is a precise concept and it does not correspond to the situation of TN1. The authors should change the way they qualify the TN1 accession.

- In the legend of Supplementary Figure 10, the authors should briefly detail what are the characteristics of the $\Delta yap1$ mutant yeast strain.

- The authors should give the complete protocol for making the metal binding assays. They refer to the protocol described in Xia et al. (2013) but there seems to be a major difference, which is that proteins are tagged and fixed to a column to collect the metal-protein complexes in Xia et al. (2013), which does not seem to be the case here since the protein is not tagged.

Reviewer #2 (Remarks to the Author):

The manuscript "Defensin-driven cadmium secretion and allocation in rice" by Luo et al., described the identification of a cysteine-rich protein in rice that facilitates the root-to-shoot mobilization of cadmium between tissues and is responsible for the difference in cadmium accumulation in leaves between two rice varieties. Overall the data are novel and interesting, the end result (less cadmium in grain) is a very important topic in the field of plant nutrition. However, the manuscript also has some inconsistencies and technical issues that need to be addressed before this manuscript can be considered for publication.

Major issues:

1. As the author mentioned, the locus was identified but the coding region for the protein itself has no SNPs, deletions or insertions; in fact, the most dramatic difference shown in Fig. 1i is that in one rice variety this gene is highly induced by cadmium but not in the other one, meaning that the phenotype observed is due to differences in the promoter region of the gene and the authors didn't discuss nor explore this fact. The difference in transcript abundance (and protein, which wasn't measured after induction) may explain the different accumulation phenotypes.

2. Despite of what the title suggests, the authors did not present any experiment or data to say that this defensin actually secretes cadmium out of the cell. So the title is misleading. In other words, the secretion of a cys-rich protein will displace the cadmium equilibrium between to compartments towards the side where the chelator is present. As it is, the authors have no evidence to say that CAL1 is actually exported with cadmium bound to it.

3. At first the results of the metal binding experiments were puzzling to me because it was hard to visualize how a cysteine rich protein purified with a 6x-His tag was unable to bind zinc and only able

to bind cadmium. By reading the Materials and Methods it is clear why... the binding buffer has 250 mM of imidazole (a metal binding molecule) and only 100 microM of the metal being investigated, that is 2500-fold difference of chelator/metal ratio so the specific activity or concentration of metals are negligible in this buffer. For these experiments to be meaningful, they have to be done at physiological conditions (apoplast/xylem-like conditions).

4. Related to the metal binding experiments, the authors do not address the issue of how CAL1 is unable to bind cadmium at pH 5.0 (the pH of the xylem) unless the signal peptide is deleted. Both, the full-length and the truncated versions have the same number of cysteines, does that mean the cadmium-binding domain is the secretion signal? Very hard to explain from the biochemical perspective but needs to be explored and explained.

Other comments:

- As mentioned earlier, the title is misleading based on the data presented.
- Introduction, it would be good to add a dollar value next to the Yuan value for comparison.
- Be consistent, if you say that China standard is 0.2 mg/Kg, use the same units for the concentration found in rice, ppm is too vague and may be expressed with different units.
- The authors found 3 QTL and only described 1 of them but the % contribution of this particular QTL to the phenotype observed is not described or explained.
- CAL1 does not "specifically bind cadmium" (line 146), this is an artifact of the experimental conditions used.

Reviewer #3 (Remarks to the Author):

Accumulation of Cd in rice and other crops grown in the presence of environmentally relevant concentrations is an important problem. Available data suggest that this accumulation is associated with health risks and potentially threatens a large number of people around the world. Understanding the mechanisms responsible for variation in Cd accumulation between plants would therefore be most welcome to tackle this problem. The manuscript presents an important advance with the mapping of a QTL contributing to leaf Cd variation in rice. It is convincingly shown that a defensin gene most likely is the causal gene. Less well supported is the proposed mechanism explaining the role of the encoded defensin in controlling leaf Cd.

Major concerns and questions:

1. I find the evidence for secretion presented in Fig. 3 not entirely convincing. Generally, such fusion protein approaches should be validated by comparison to markers of known localization. Specifically, I had problems interpreting the result for mRFP-CAL1 in 3g: why is the pattern apparently different from the mRFP control? Why does the manuscript text refer to a "ubiquitous expression pattern"?
2. Secretion should be further supported by independent evidence. LC-MS detection of defensin-derived peptides is possible. This should be employed systematically to detect the protein in the apoplast of expressing tissues. In my view this would be a straightforward experiment.
3. The detection of the fusion protein in xylem exudates (better term than xylem sap) lacks the proper

control in my view. One should compare to mRFP-CAL1 that is supposedly not secreted. The key is not detection of the protein at all but the compartment it is found in. Thus, the control should be a line expressing the protein presumably in the wrong place, i.e. a non-secreted form.

4. Secretion of a defensin-Cd complex as the underlying mechanism is difficult to comprehend. More data on the metal-binding is needed. Secretion would involve passage through reducing and oxidizing cellular compartments. How can the Cd binding work in both? The biochemical characterization should definitely include experiments with and without reducing agents.

5. A second point is the stoichiometry. How many Cd atoms are bound per protein molecule? Does that number fit the number of cysteine residues at least principally?

6. What are the defensin amounts in the xylem in relation to the Cd concentration? What could possibly be the contribution of defensin to the Cd translocation as a binding partner? If the Cd/defensin ratio is say 1000 this could not work. Hence, such numbers are important and it should be possible to obtain at least informative estimates.

7. Has site-directed mutagenesis been tried on the cysteines to see possible effects on Cd binding?

8. What is the Cd accumulation like in yeast and E. coli? Is it consistent with Cd binding by defensin?

9. Xylem exudate purity is a challenge. It would therefore be important to present data for other elements as well. I found data only for the tissues.

10. There is no discussion of existing knowledge about defensins; no classification of CAL1; no discussion specifically of alternative hypotheses as to the metal-related activity of defensins, e.g. the work implicated defensins in Zn tolerance.

More specific and minor points:

11. In Fig. 2: differences between TN and CJ06 appear to be much bigger for Cd in leaves and Cd in xylem sap than in Fig. 1. Is that correct and what is the reason?

12. Why are there no xylem exudate data for mutants in TN1 background (Fig. 2)? Cd in leaves is shown.

13. Fig. 4 b,c: is the statistical analysis appropriate? At the 0 h time point there are already differences; thus, it is not sufficient to simply compare the 6 h values.

14. What is meant by "series secretion protein" (l. 122)?

15. One should not refer to "complemented the Cd accumulation phenotype in CJ06" (l. 79); this sounds like a mutant phenotype which it is not.

16. The authors refer in the introduction to the "safe threshold" for Cd. This is not entirely correct. It is important to emphasize that any threshold is only "provisionally safe".

Reviewer #1 (Remarks to the Author):

“From an analysis of a collection of rice accessions, the authors identified one accession (TN1) that accumulated high levels of Cd both in its leaves and grains.....The data reported here are for sure novel. They will be of large interest because, besides interesting the research community working on metal transport and accumulation in plants, they point a role of defensin-like proteins in a process in which these proteins were up to now not expected to play a role (at least to my knowledge). As such, these results should contribute to bring new perspectives and make thinking evolve in different fields of research. In addition, these results show that it could be possible to both increase the accumulation of cadmium in a plant while ‘preserving’ the grain, and thus this work should lead to interesting practical outputs.

We thank the reviewer for his/her positive feedback and totally agree that this work would be of large interest.

Major issues:

1. In a large part because the authors made efforts to bring different pieces of evidence to support their conclusions the work is as a whole convincing. Although, specific points should be improved. The genetic part and the demonstration that CAL1 is responsible for the phenotype of interest both appear clear. The localization of the expression of the gene seems clear, although the authors only exploit the fact that the gene is expressed in the xylem parenchyma in roots and not the fact that it is also expressed in other tissues. Concerning the analysis of the protein, I have a major concern about the western analyses (Figure 4). To my knowledge, the mRFP alone is already heavier than 30 kDa. When it is fused to the defensin-like protein, it should be at least 5 kDa heavier. This is not compatible with the results presented in Figure 4. I thus wonder whether the authors indeed analyzed the defensin-like protein or a cleaved mRFP (this occurred at least once in my lab) on their western blots. Thus this may question the actual subcellular location of the protein since it is detected only through the analysis of the mRFP.

The mRFP we used here is ~25 kDa (Huang et al., 2014, Plant Cell, 26, 2505–2523), not 30 kDa. So with the 5kDa mature CAL1, the size of the CAL1-mRFP fusion (without the secretion signal peptide) is ~30 kDa as we had reported.

Moreover, the CAL1 fragment was firstly detected in xylem sap by direct ESI-MS analysis (Supplementary Fig. 14), which drove us to then use Western blot to confirm the ESI-MS assay data (Fig 4), and these two lines of evidences consistently support the conclusion that CAL1 presents in xylem sap.

The mRFP protein sequence is listed below:
MASSDVIKEFMRFKVRMEGSVNGHEFEIEGEGEGRPYEGTQTAKLKVTKGG
PLPFAWDILSPQFQYGSKAYVKHPADIPDYLKLSFPEGFKWERVMNFEDGGV
VTVTQDSSLQDGEFIYKVKLRGTNFPDGPVMQKKTMGWEASTERMYPEDG
ALKGEIKMRLKLDGGHYDAEVKTTYMAKKPVQLPGAYKTDIKLDITSHNE
DYTIVEQYERAEGRHSTGA

2. I am not familiar with protein-metal binding assays, but I feel that the authors should improve the quality of the data or of the presentation. The protocol of the in vitro binding assay should be given entirely (in supplementary material if it is too long). In my opinion, the experiment cannot be reproduced, as it stands. The quantity of Cd that is bound to the protein should be expressed in a mol/mol ratio instead of in a mass/mass ratio. Indeed, if the mol/mol ratio is far below 1 or higher than let's say 2, this should question the significance of the results. I am not sure that the 'tagging protein' is a good negative control. I think that any cystein-rich protein would bind cadmium as CAL1 does, wouldn't it? The protoplasts assays should be done after equivalent durations of incubation (see details below). The positive point that I note is the metal specificity of the 'binding' but this may be a bit weak.

We agree with the reviewer that the quality of the binding assay and of the presentation could be improved. Indeed, our previous description of the protocol had resulted in confusion to other reviewers. We have done the following to address the reviewer's concern:

- 1) We improved the protocol, included the new data (Fig. 4a, Supplementary Fig.11b) and added the information into the main text (line 357-362);
- 2) We adopted the widely used ITC (isothermal titration calorimetry) assay to further demonstrate the Cd binding activity of CAL1, and the new data was also included (Supplementary Figs 10, 12, Supple. Table 4) and described in the main text (lines 164-165);
- 3) The quantity of Cd bound to the protein was now expressed in a mol/mol ratio (Fig 4a, Supplementary Figures 11, 16b), which showed values around 1.2-1.5.
- 4) The protoplast assays were also revised as suggested (Fig. 4c). Specifically, we included the 6h result for the mutant protoplasts to allow same incubation time for both, which was actually comparable to the previous 12 h result.

3. *I am also not really convinced by the fact that the TN1 and CJ06 rice accessions display the same ability to uptake Cd (Supplementary Figure 1). Is the assay really measuring the Cd uptake? Indeed, if TN1 translocates more Cd from the roots to the shoot than CJ06 while having the same concentration of Cd in the roots (Supplementary Figure 5), it should take up more Cd than CJ30. This should be commented in greater details.*

The Cd uptake assay was performed according to published protocol as described (Ueno et al., 2009, *New Phytologist*, 182, 644-). In terms of why TN1 didn't take up apparently more Cd than CJ06 when more Cd was translocated to shoots but similar Cd concentrations were observed in roots, we suspected that the Cd concentration in roots (supplementary Figs. 6 and 7) was far higher than in shoots (Fig 1), thus the enhanced translocation didn't apparently affect Cd concentration in roots. The hypothesis was supported by similar observations in several other studies on either heavy metals or mineral nutrients (Takahashi et al., 2012, *Plant Cell & Environment*, 35(11), 1948- ; Satoh-Nagasawa et al., 2012, *Plant Cell Physiol.* 53(1): 213-; Ueno et al., 2011, *Journal of Experimental Botany*, 62, 2265-.)

To address this concern, we have rephrased the sentence to more neutrally interpret our data (lines 70, 72).

4. *The authors found that a defensin-like protein is involved in the transport and accumulation of cadmium. But what is a defensin? Why do the authors not give a single word about defensins?*

In the revised manuscript, a brief introduction about defensins has been included (lines 46-51). We didn't mention defensins in the previous manuscript only because we worried too much about the length of introduction part.

5. *Why do the author name their protein 'defensin' although they have not checked*

that CAL1 indeed has a defensin activity. They have to use the terms 'defensin-like protein' instead of 'defensin'.

We have improved the terms as suggested (in the title and also other places when necessary).

6. At first glance, the result may appear surprising. One could have expected transporters or regulators to have been involved. It is surprising that the authors do not comment about the fact that a defensin-like protein would play the role that they identified. I made a rapid bibliography survey and found that plant and animal defensins have actually already been described as being related to metals (metal tolerance, metal binding ...). I think that it is necessary that the authors connect their findings with data already published showing that defensins are involved in 'metal physiology'.

To address this concern, we have cited these literatures and comment briefly (lines 51-55).

Previous studies focused on defensins' contribution to innate immune response and antifungal activity. There were several recent literatures reporting defensin's role in metal tolerance (Mirouze et al., 2006, Plant Journal 47, 329-; Shahzad et al., 2013, New Phytologist, 200, 820-) and also in metal binding (Zhang et al., 2013, ACS Chem Biol, 8(9), 1907-), but how it works remains to be investigated.

7. I wonder why the authors screened the rice collection for variants in the coding sequence (which is not a bad idea per se) and not in the promoter (or intron) to catch similar mutations to those found in TN1 and that would be responsible for the Cd-induction of the expression and the high cadmium accumulation. Is TN1 the only rice accession harbouring mutations in the CAL1 promoter ? What is the phenotype of other accessions showing the same variations in their promoter sequences ?

We understand the reviewer's concern. However, given a bunch of mutations occurred in *CAL1* promoter sequences between TN1 and CJ06, it is really difficult (unaffordable) to figure out which mutation(s) contribute to the enhanced *CAL1* expression in TN1. So we instead tried to find other *CAL1* haplotypes with enhanced expression levels and mutations in their promoter regions, but the elevated expression and mutations did not always correlate to elevated Cd accumulation, which is normal considering that *CAL1* is only one of those quantitative trait loci and the genetic background for different accessions are highly variant. Interestingly, we did find a natural variation in the coding sequence of *CAL1*, as reported in our manuscript (Fig. 4f) and noticed by the reviewer, which is highly conserved and correlates with the Cd accumulation levels in different rice accessions, although they have different genetic background.

To address this concern, we have added some text to better describe the rationale for our experimental design (lines 203-209).

More specific issues:

8. 47-48: It is necessary that the authors specify which core collection they screened. They referred to a paper from Zhang et al. (2011) but this is not sufficient since (1) several different core collections are presented in that paper and (2) no "China Rice Core Collection" is defined in that paper. The authors thus have to mention whether they screened the 932 accessions constituting the Core Collection, or the 189 accessions constituting the mini Core Collection, or any other number of accessions.

Sorry for this confusion, the rice varieties we used include the mini core collection and several other rice accessions which are thought important or have been crossed to other accessions, thus corresponding genetic populations are readily available. The term China Rice Core Collection is not accurate and we have

improved our wording to more precisely describe the rice materials we used (lines 61-62).

9. 47-48: It is absolutely necessary that the authors give a Supplementary Table with the Cd, Fe, Zn, Mn and Cu grain and leaf content measurements in all the accessions that they screened.

We understand the reviewer's interest about the dataset, and have included the profile of Cd accumulation in grains of the rice accessions we used (Supplementary Fig 1a, Supplementary Table 1), which we believe should have been included. We also included other metals as required (Supplementary Fig 1b, Supplementary Table 1), although those data seem not so relevant to the current research, but they would benefit other colleagues who might be interested. Ionic screening was not performed for rice leaves, and we profiled metal accumulation there only after target accessions were identified, which helped to save money.

10. 51: the authors should mention in the text the relative difference between Cd content in grain and leaves of the TN1 cultivar compared to the CJ06 cultivar. The reader should not have to go to Figure 1 to see the difference, which is actually not quantified in that Figure.

Thanks for pointing this out, we have improved the text as suggested (line 65).The difference is 3 fold or so.

11. 58: since the paper describing the doubled haploid population is not easily available, I ask the authors to mention the number of doubled haploids that they used for their QTL approach. I also ask them to mention how many markers they used for the QTL mapping.

We actually had described the DH population and the genetic map in the Method section by referring to a published paper. However, to allow easier reading, we now added the detailed information (119 DH lines, and 177 SSR markers for rough QTL mapping) in the text (lines 250-251), as suggested.

12. 143, as well as in the legends of Figure 4 (l. 536), of Supplementary Figure 9 and of Supplementary Figure 13: what is the (control) 'tagging protein' ?

The tagging protein is TF (trigger factor), which is a commercial tag protein regularly used to fuse to target proteins. To allow easier reading, we improved the text (line 161).

13. In Figure 4, I wonder why the authors have used a 6-hour-long incubation time before evaluating the Cd content in protoplasts of the complementation lines, while they used a 12-hour-long incubation time for the protoplasts of the mutant lines. Considering that they observe a difference at the end of the 6-hour-long incubation time but not at the end of the 12-hour-long incubation time, is it really the different origins of the protoplasts that makes the difference (as the author claim; l. 152-154) or the length of the incubation time? I consider that the authors should show data obtained at the end of the same length of incubation time.

We thank the reviewer for his/her careful reading. It is normal to use different incubation time because the protoplasts were from different genetic background, thus requiring different conditions. Specifically in our case, one data derived from complementation lines representing gain-of-function, which is dominant and cannot easily be affected by other redundant gene(s); the other one derived from mutant lines representing loss-of function, whose effect is frequently observed to be totally or partially affected by functional redundancy from other genes. So based on our pilot experiment, we had showed the longer incubation (12h) result for the mutant lines. It

is worth noting that incubation time for different dataset won't affect the final conclusion, as those data were compared to their own control respectively.

As required, we now included the mutant data at the 6h time point to allow same incubation time, and the difference is still significant (Figure 4c).

14. 158-159 and Legend of Figure 4: it is absolutely necessary that the authors mention in the legend of Figure 4 that they used an anti-RFP antibody to check the presence of the CAL1::mRFP fusion protein in the xylem sap or the leaf blade and sheath.

We have added the information as suggested (highlighted in the legend of Figure 4, line 601).

15. In Figure 4, it would be much better to express the Cd content supposedly chelated to the defensin-like protein in mol / mol (instead of in mass / mass).

As suggested, we have reformatted the dataset to express them in mol/mol ratio, which we agree is much better.

16. 162-167: the English should be improved. I am not sure to have properly understood the point raised by the authors.

We have improved the English with the help from an English editing company. We also carefully revised the manuscript to better describe our points and to clarify possible confusions.

17. Legend of Supplementary Figure 1: Is it the whole plants that are maintained at 4°C, or the root systems only ?

We exposed the whole plants to 4°C or 28°C according to the method as described in the cited reference (Ueno et al., 2009, *New Phytologist*, 182, 644-).

18. Legend of Supplementary Figure 3: TN1 is NOT a Cd hyper-accumulating accession. The concept of hyper-accumulation is a precise concept and it does not correspond to the situation of TN1. The authors should change the way they qualify the TN1 accession.

This typing error has been corrected (Legend of Supplementary Figure 4).

19. In the legend of Supplementary Figure 10, the authors should briefly detail what are the characteristics of the $\Delta yap1$ mutant yeast strain.

Brief information of the $\Delta yap1$ mutant was added as required (legend of Supplementary Figure 13). $\Delta yap1$ mutant was impaired in activating YCF1 expression, thus becoming highly sensitive to Cd.

20. The authors should give the complete protocol for making the metal binding assays. They refer to the protocol described in Xia et al. (2013) but there seems to be a major difference, which is that proteins are tagged and fixed to a column to collect the metal-protein complexes in Xia et al. (2013), which does not seem to be the case here since the protein is not tagged.

We thank the reviewer for pointing this out. Our method indeed had been essentially improved/simplified.

To address the concern, we have detailed the improved protocol to better describe how we performed the metal binding assay (lines 357-362).

Reviewer #2 (Remarks to the Author):

The manuscript “Defensin-driven cadmium secretion and allocation in rice” by Luo et al., described the identification of a cysteine-rich protein in rice that facilitates the root-to-shoot mobilization of cadmium between tissues and is responsible for the difference in cadmium accumulation in leaves between two rice varieties. Overall the data are novel and interesting, the end result (less cadmium in grain) is a very important topic in the field of plant nutrition. However, the manuscript also has some inconsistencies and technical issues that need to be addressed before this manuscript can be considered for publication.

We thank the reviewer for his/her recognition of our research, and have improved the manuscript as detailed in the response to reviewers.

Major issues:

1. As the author mentioned, the locus was identified but the coding region for the protein itself has no SNPs, deletions or insertions; in fact, the most dramatic difference shown in Fig. 1i is that in one rice variety this gene is highly induced by cadmium but not in the other one, meaning that the phenotype observed is due to differences in the promoter region of the gene and the authors didn't discuss nor explore this fact. The difference in transcript abundance (and protein, which wasn't measured after induction) may explain the different accumulation phenotypes.

We agree with the reviewer that difference in transcript abundance may explain the different accumulation phenotypes. Moreover, we found that variation in protein sequence also contributes to different Cd accumulation (Fig. 4g, h). Both of these phenotypes are expectable.

As suggested also by reviewer #1, we have added more sentences to better describe our results and to clarify the possible confusion (please refer to the response to comment #7 by reviewer #1, and also to the textual modification in lines 203-209)

2. Despite of what the title suggests, the authors did not present any experiment or data to say that this defensin actually secretes cadmium out of the cell. So the title is misleading. In other words, the secretion of a cys-rich protein will displace the cadmium equilibrium between to compartments towards the side where the chelator is present. As it is, the authors have no evidence to say that CAL1 is actually exported with cadmium bound to it.

We understand the reviewer's concern, however, we made such a conclusion basing on the following reasons/evidence:

1) CAL1 is synthesized in the cytosol. It is demonstrated to bind Cd in cytosol and extracellular space. CAL1 is secreted from cytosol to extracellular space. These observations suggest that CAL1 should be exported with Cd bound to it, because otherwise, plants have to develop some machinery to detach Cd from CAL1 before exportation out of cells and then chelate it again in extracellular space, which is not energy efficient and unreasonable.

Phytochelatins (PCs) from most recent research (Song et al., 2010, PNAS, 107, 21187-; Park et al., 2012, Plant J, 69, 278-) and other bunch of literatures could provide good examples helping understand the CAL1-driven Cd secretion. PCs are also small cysteine-rich proteins (providing thiols). Regularly one Cd binds to 2-4 thiols from PCs to form the PCs:Cd complex (Hirata et al., 2005, J Bioscience and Bioengineering, 100: 593-), which was then transported across tonoplast (vacuole membrane) by the transporters ABCC1 or HMT1 in plants or yeast. Though in our case, we prefer another vesicular trafficking pathway, as CAL1 has a typical secretion signal peptide at its N-terminus. We added some text to discuss the possible underlying mechanisms (lines 231-232).

2) Further evidence came from the observations that Cd levels decreased faster in N47s' protoplasts than its control CJ06 (Fig. 4b). Consistently, slower decline in Cd concentration was observed in the function-disrupted *cal1* mutant plants (*cal1*-T2 and *cal1*-A, Fig. 4c).

3) The third line of evidence: ectopic expression of CAL1 in yeast and E coli enhanced metal tolerance and accumulation in cells (Supplementary Fig. 13). The former (enhanced metal tolerance) demonstrated chelation and hence detoxification of Cd, while the latter (enhanced metal accumulation in cells) demonstrated that without the secretion signal peptide, CAL1 could not be secreted thus Cd could not either.

These multiple lines of evidence suggest that our conclusion is evidence-based. We are also trying to get direct evidence (like direct detection of the complex translocating across the plasma membrane), but incapable only because of technology limitation. However, the evidences we provided already consistently suggest that CAL1 is exported with Cd bound to it. In the current version, we added some information about defensin (lines 46-55), which we believe would help clarifying the confusion. Also we have rephrased the sentences to soften our statements (lines 170-171, 181)

3. At first the results of the metal binding experiments were puzzling to me because it was hard to visualize how a cysteine rich protein purified with a 6x-His tag was unable to bind zinc and only able to bind cadmium. By reading the Materials and Methods it is clear why... the binding buffer has 250 mM of imidazole (a metal binding molecule) and only 100 microM of the metal being investigated, that is 2500-fold difference of chelator/metal ratio so the specific activity or concentration of metals are negligible in this buffer. For these experiments to be meaningful, they have to be done at physiological conditions (apoplast/xylem-like conditions).

We agree with the reviewer that imidazole in the washing buffer might interfere with the metal binding, however, considering Cd and Zn binding were performed under the same condition (100 uM metal and 250 mM imidazole), and Cd was detected to be bound to CAL1, we tend to believe that the effect by imidazole might not be as significant as the concentration itself might indicate.

To address the reviewer's concern, we improved our previous analysis by replacing the imidazole buffer, and further adopted another technology (ITC assay) to

test the metal binding. As expected, all these newly obtained data consistently support our previous conclusion (Fig 4a, and Supplementary Figs. 10, 11, 12, 16b). We also revised the method part accordingly (lines 357-362).

4. Related to the metal binding experiments, the authors do not address the issue of how CAL1 is unable to bind cadmium at pH 5.0 (the pH of the xylem) unless the signal peptide is deleted. Both, the full-length and the truncated versions have the same number of cysteines, does that mean the cadmium-binding domain is the secretion signal? Very hard to explain from the biochemical perspective but needs to be explored and explained.

We used different pH to simulate the acidity environment in cytosol and xylem respectively, which did suggest that the mature CAL1 (without secretion signal) might bind Cd efficiently in both conditions, while the full length one can do so apparently at pH 7.5. These results are consistent to our postulation that mature protein can bind Cd in both cytosol and xylem sap, while the full length one does not.

We are also very interested in the underlying mechanism, and have started a new project to investigate the 3D structure of this protein, which we believe would answer the specific question as well as other questions, such as the metal binding specificity as also noticed by reviewers.

To address the reviewer's concern, we added some sentences to discuss that acidic pH might interact with the secretion signal peptide to affect CAL1's binding ability to Cd (lines 162-163), though this is not supposed to happen in real world, as only mature protein is secreted and undergoes long-distance transport in xylem.

Other comments:

5. As mentioned earlier, the title is misleading based on the data presented.

We have modified the title correspondingly, given the discussed reasons in the response to comment #2, and also taken into account the concern by reviewer #1.

6. Introduction, it would be good to add a dollar value next to the Yuan value for comparison.

We have added the required information in the text (line 35). According to the exchange rate in 1996, 20 billion Yuan amounts to 2.4 billion US dollars.

7. Be consistent, if you say that China standard is 0.2 mg/Kg , use the same units for the concentration found in rice, ppm is too vague and may be expressed with different units.

We have revised the text as suggested (line38).

8. The authors found 3 QTL and only described 1 of them but the % contribution of this particular QTL to the phenotype observed is not described or explained.

To allow easier reading, we added the required information (in Supplemental Table 2) to the main text (line 76)

9. CAL1 does not “specifically bind cadmium’ (line 146), this is an artifact of the experimental conditions used.

As indicated by our new data obtained from the improved experimental design and also from the newly added ITC analysis, CAL1 most likely binds specifically to Cd. Please refer to our detailed response to comment #3.

To address the concern, we have reworded the text to soften our statement (lines 170, 228-229).

Reviewer #3 (Remarks to the Author):

Accumulation of Cd in rice and other crops grown in the presence of environmentally relevant concentrations is an important problem. Available data suggest that this accumulation is associated with health risks and potentially threatens a large number of people around the world. Understanding the mechanisms responsible for variation in Cd accumulation between plants would therefore be most welcome to tackle this problem. The manuscript presents an important advance with the mapping of a QTL contributing to leaf Cd variation in rice. It is convincingly shown that a defensin gene most likely is the causal gene. Less well supported is the proposed mechanism explaining the role of the encoded defensin in controlling leaf Cd.

We thank the reviewer for his/her comment, and all these comments by reviewers have helped us to improve our work.

Major concerns and questions:

1. I find the evidence for secretion presented in Fig. 3 not entirely convincing. Generally, such fusion protein approaches should be validated by comparison to markers of known localization. Specifically, I had problems interpreting the result for mRFP-CAL1 in 3g: why is the pattern apparently different from the mRFP control? Why does the manuscript text refer to a “ubiquitous expression pattern”?

Indeed the pattern in Fig 3g looks a little bit different from the mRFP control in Fig.3e, which we believe might be largely due to the fact that those cells were at different plasmolysis stages. However, it is still not hard to tell that the fluorescence was observed in nuclei and cytoplasm of both cells, which is quite typical for this kind of proteins. In terms of the “ubiquitous expression pattern”, we agree it is not accurate

or even kind of misleading, and have revised the text to better describe our result (lines 143-144).

2. Secretion should be further supported by independent evidence. LC-MS detection of defensin-derived peptides is possible. This should be employed systematically to detect the protein in the apoplast of expressing tissues. In my view this would be a straightforward experiment.

We agree that LC-MS detection of defensin-derived peptide is possible and we actually did a similar analysis using ESI-MS, which directly detected the CAL1 fragment (Supplementary Fig.14).

Also secretion was indeed supported by independent evidence. Specifically, we made such a conclusion basing on three lines of evidence: 1) subcellular localization using mRFP (Fig.3); 2) Western blot analysis (Fig. 4d); 3) ESI-MS detection of CAL1 fragment (Supplementary Fig 14). All those data consistently demonstrated that CAL1 was secreted to apoplastic compartment. Regarding LC-MS, we haven't established the method to detect defensin-derived peptides, but the ESI-MS platform is mature enough to do the same thing.

3. The detection of the fusion protein in xylem exudates (better term than xylem sap) lacks the proper control in my view. One should compare to mRFP-CAL1 that is supposedly not secreted. The key is not detection of the protein at all but the compartment it is found in. Thus, the control should be a line expressing the protein presumably in the wrong place, i.e. a non-secreted form.

We agree with the reviewer that it would be even better to use mRFP-CAL1 as a control. However, given three lines of evidence had been collected to support our conclusion that CAL1 was secreted into apoplastic compartments, and both the Western blot (Fig. 4d) and ESI-MS assay (Supplementary Figure 14) had directly detected CAL1 in xylem exudates (please also refer to the response to the above

comment #2), we were unable to find a more convincing reason to make an opposite conclusion.

4. Secretion of a defensin-Cd complex as the underlying mechanism is difficult to comprehend. More data on the metal-binding is needed. Secretion would involve passage through reducing and oxidizing cellular compartments. How can the Cd binding work in both? The biochemical characterization should definitely include experiments with and without reducing agents.

We appreciate the reviewer's suggestion, and have adopted additional ITC analysis to determine metal binding, which together with another line of evidence (with improved assay protocol) suggested that CAL1 binds to Cd.

In terms of the underlying mechanism for secretion of the defensin-Cd complex, we have described in detail in the response to comment # 2 by reviewer #2, and made corresponding revision (lines 231-232). Briefly, phytochelatins (PCs) might provide a good example to help understanding how this works: like CAL1, PCs are also small cysteine-rich proteins, in which the thiols bind to Cd to form the PCs:Cd complex (Cd/thiols ranges from 1:2-4), and the complex was then transported across tonoplast (vacuole membrane) by the transporters ABCC1 or HMT1 in plants or yeast. So basically, Cd is coated by these kinds of Cys-rich proteins, and the complex is electrically neutral, thus "reducing and oxidizing cellular compartments" is not required.

The CAL1:Cd complex might be secreted via a similar way as used by PCs:Cd, or more possibly through a vesicular trafficking pathway, as CAL1 has a typical secretion signal peptide at its N-terminus. My lab is collaborating with a cell biology lab to find out the exact answer, which if lucky enough, we wish could be reported in a separate paper in future.

5. A second point is the stoichiometry. How many Cd atoms are bound per protein molecule? Does that number fit the number of cysteine residues at least principally?

There are 8 cysteine residues in each CAL1 molecule (please refer to the newly added information about this protein, lines 46-55), but our data indicated that only three of them significantly affected the Cd-binding ability (supplementary Fig. 11, Supplementary Table 4), suggesting the other 5 residues might function to stabilize/shape the specific 3D structure of this protein. Our Cd-binding assay was consistent to this hypothesis, as a molar ratio of Cd/CAL1 was repeatedly observed near 1.5 (Fig. 4a, Supplementary Fig. 11b). Taking into account possible background signal (molar ratio for the control protein TF), we postulated a most possible stoichiometry of 1 for Cd/CAL1 (by molar ratio), which fits to the numbers of functional thiols. Further structural assay is in progress in my lab, which may provide a more accurate model and answer many unanswered questions, including the mentioned metal binding specificity.

6. What are the defensin amounts in the xylem in relation to the Cd concentration? What could possibly be the contribution of defensin to the Cd translocation as a binding partner? If the Cd/defensin ratio is say 1000 this could not work. Hence, such numbers are important and it should be possible to obtain at least informative estimates.

We didn't really estimate the defensin amount due to technical difficulties. However, given ~20% decrease of Cd was observed in xylem of the *cal1* mutant compared with the wild type control, we postulated that defensin might contribute to ~ 20% of the Cd translocation rate, which is reasonable as this QTL was estimated to contribute ~13% of the overall variance (Supplementary Table 2). Given this kind of estimation was pretty difficult and rough, we prefer not to over-interpret our data.

7. Has site-directed mutagenesis been tried on the cysteines to see possible effects on Cd binding?

This is a great comment. As discussed in response to the above comment # 5, we did do such an experiment shortly after we submitted the paper, and have included the data in the current version (Supplementary Fig. 11, Supplementary Table 4, lines 166-168), which showed only 3 of them significantly affected Cd binding. It is worth noting that Cd binding capacity was affected not only by the thiols, but also by other factors such as the mutation at site 70 along the protein sequence (Fig. 4f).

8. What is the Cd accumulation like in yeast and E. coli? Is it consistent with Cd binding by defensin?

Yes, it is consistent and Cd accumulation increased as expected, because without the secretion signal peptide, CAL1 could not be secreted to extracellular compartments. These data further suggest that CAL1 binds Cd, and secretion of the complex promotes Cd efflux. We have included the data (Supplementary Figure 13b, d)

9. Xylem exudate purity is a challenge. It would therefore be important to present data for other elements as well. I found data only for the tissues.

Thanks for pointing this out, we have included the data for other elements (Supplementary Figure 6c, lines 120-121), which indicate that except for Cd, concentrations of other analyzed metals was not affected.

10. There is no discussion of existing knowledge about defensins; no classification of CAL1; no discussion specifically of alternative hypotheses as to the metal-related activity of defensins, e.g. the work implicated defensins in Zn tolerance.

We have included more information (lines 46-51) and discussed (lines 51-55) as also suggested by other reviewers (please refer to comments # 4&6 by reviewer #1).

More specific and minor points:

11. In Fig. 2: differences between TN and CJ06 appear to be much bigger for Cd in leaves and Cd in xylem sap than in Fig. 1. Is that correct and what is the reason?

We are not sure if we get the reviewer's point right, but the data (Fig 1c,d vs Fig. 2a,b) indicate the opposite: the difference in Fig.1 is bigger than in Fig. 2, because in Fig 1, TN1 and CJ06 are the original rice accessions, which contain all the QTL genes that contribute to Cd over-accumulation in TN1, while in Fig. 2, the NIL(TN1) and NIL(CJ06) are near isogenic lines (NIL) that are different only in the genetic region containing the *CAL1* locus. If the reviewer meant the absolute Cd concentration, it is normal that the values will fluctuate from batches to batches, because metal accumulation is quantitative trait and strongly affected by environments, thus we only compare data from the same batch.

12. Why are there no xylem exudate data for mutants in TN1 background (Fig. 2)? Cd in leaves is shown.

Thanks for reminding. The mentioned data is included in the current version (Supplementary Figure 7a, lines 123-124). We have so many genetic materials that it did occur to us to ignore something here and there.

13. Fig. 4 b,c: is the statistical analysis appropriate? At the 0 h time point there are already differences; thus, it is not sufficient to simply compare the 6 h values.

In the current version, the ratio of 6h/0h instead of each data was compared between the wild type and the mutant/transgenic plants.

14. What is meant by "series secretion protein" (l. 122)?

To clarify this confusion, we have revised the text accordingly (line 139). Defensins consist of a large family, and most of the family members share a similar N terminal signal peptide and a C terminal cysteine-rich domain, thus we had called them “series secretion protein”.

15. One should not refer to “complemented the Cd accumulation phenotype in CJ06” (l. 79); this sounds like a mutant phenotype which it is not.

The term “complement” is actually used regularly in QTL-related research, which is probably due to the fact that these natural variations represent another kind of “mutants”. However, we revised the text accordingly as suggested (line 95).

16. The authors refer in the introduction to the “safe threshold” for Cd. This is not entirely correct. It is important to emphasize that any threshold is only “provisionally safe”.

As suggested, the “safe threshold” for Cd in the introduction had been revised as “provisionally safe threshold” (line 39).

In summary, we addressed all the concerns by including more experimental data or improving the text to better describe our results and to clarify our points.

Reviewers' comments:

Reviewer #1 (Remarks to the Author):

Following careful reading of the answers that the authors made to my concerns (reviewer 1), as well as to the changes that they made to the manuscript accordingly, following also a complete re-reading of the revised version of the manuscript, I consider that the authors made all the requested changes appropriately. I did not detect any problem or difficulty. Thus I fully support the publication of the revised version of the manuscript.

Reviewer #2 (Remarks to the Author):

This is a re-submission and a very improved version compared to the original manuscript. I have read ALL the reviewers' concerns and the authors' responses and I can say that all concerns have been addressed properly and the manuscript has been modified to reflect such changes. Therefore, I support the publication of this ms in its present form.

Reviewer #3 (Remarks to the Author):

I acknowledge improvements through the inclusion of more data. Unfortunately, however, my concerns have only partially been addressed in the revised version.

Comment 2 - Apparently no attempt was made to detect the protein in the apoplast of expressing tissues

Comment 3 - Xylem sap analysis can be confounded by partial lysis of cells. Thus, a good control would be the analysis of plants expressing a non-secreted form of CAL1. In my view this is essential.

Comment 4 -

a. I am still not convinced by the binding assays. The authors state "the Cd binding capacity for the full length CAL1 decreased essentially to a level close to the control tagging protein". I would assume the difference is still highly significant when tested.

b. The bars of new Fig. 4a resemble those of the previous Fig. 4a even though the assay conditions were changed quite a bit. However, if I am not mistaken the molar ratios calculated from previous Fig. 4a are about factor 10 different from the ratios shown in new Fig. 4a.

c. the oxidation/reduction question is disregarded even though this should have a major influence on metal binding as can be expected for a protein with disulfide bridges and as shown, for example, with a human defensin in Zhang et al ACS Chem Biol 2013.

Comment 6 - I do not see why it is not possible to derive an estimate of CAL1 protein level when the fusion protein can be detected in xylem sap with an anti-RFP antibody. The CAL1/Cd stoichiometry in the xylem sap is a crucial piece of information. As a side note: the apparent lack of a change in fusion protein levels in xylem sap in Cd treated plants (Fig. 4d) is somewhat inconsistent with the transcript data for roots.

Comment 11 - Regarding my comment on variations in Cd levels across experiments: between Figs. 2f and 2g there is a difference of 10fold between the leaf values for two rice accessions; in another comparison TN1 seedlings accumulate around 20 $\mu\text{g/g}$ Cd when exposed to 10 μM Cd for 7 days

according to Fig. 1c while in Fig. 2f the value is around 100. These differences cannot be explained by batch-to-batch variation in my view. Rather, they point to possible technical problems.

We have included new data and revised our manuscript to fully address the remaining concerns, which were highlighted in the main text of our manuscript.

Response to reviewers

Reviewer #1: --*Following careful reading of the answers that the authors made to my concerns (reviewer 1), as well as to the changes that they made to the manuscript accordingly, following also a complete re-reading of the revised version of the manuscript, I consider that the authors made all the requested changes appropriately. I did not detect any problem or difficulty. Thus I fully support the publication of the revised version of the manuscript.*

We thank the reviewer for his/her comments, which greatly helped to strengthen our manuscript.

Reviewer #2: --*This is a re-submission and a very improved version compared to the original manuscript. I have read ALL the reviewers' concerns and the authors' responses and I can say that all concerns have been addressed properly and the manuscript has been modified to reflect such changes. Therefore, I support the publication of this ms in its present form.*

We thank the reviewer for his/her time and patience to read all the concerns, and his/her great comments have helped greatly to improve our manuscript.

Reviewer #3:

-- *“Comment 2 - Apparently no attempt was made to detect the protein in the apoplast of expressing tissues”*

The reviewer suggested using mass spectrometry to detect CAL1 protein in the apoplast (original comment: *“Secretion should be further supported by independent evidence. LC-MS detection of defensin-derived peptides is possible.....straightforward experiment”*). We **DID perform** such an experiment and have already included the data in both submissions (**Supplementary Fig. 15, lines 186-187**), and these data together with other lines of evidence (*subcellular localization in Fig. 3c-h, Western blot in Fig. 4d, and also the newly identified presence in guttation fluid of Suppl. Fig. 16*) consistently suggest that the CAL1 protein is present in the apoplast.

--*Comment 3 - Xylem sap analysis can be confounded by partial lysis of cells. Thus, a good control would be the analysis of plants expressing a non-secreted form of CAL1. In my view this is essential.*

It will regularly take at least 10 months generating transgenic rice plants to express the non-secreted form of CAL1 as a control. **To more efficiently address the reviewer's concern, we now performed two more experiments as also suggested by the editor: 1) expressing the non-secreted form of CAL1 in yeast; 2) collecting rice guttation drops** (i.e. the exudation of liquid from the surface of intact leaves through the hydathode, which can provide us xylem exudates without injuring the plant tissues).

Consistently, as shown in **Supplementary Fig. 14**, Cd exudation into the growth media was significantly enhanced by the full length CAL1, but not by the non-secreted form (mRFP-CAL1), providing further support to our hypothesis. We added more sentences to describe our new data (**lines 182-184**).

Moreover, as shown in **Supplementary Fig. 16**, the guttation data clearly shows the presence of CAL1 protein in the guttation fluid (**lines 189-190**). Please note that the other three lines of evidence (*subcellular localization in Fig. 3c-h, and Western blot in Fig. 4d, MS detection of CAL1-derived peptide in Supplementary Fig. 14*) together have demonstrated that CAL1 is secreted to apoplast, and this additional line of evidence further addressed the concern on potential pollution from partial cell lysis.

--Comment 4 -

a. I am still not convinced by the binding assays. The authors state “the Cd binding capacity for the full length CAL1 decreased essentially to a level close to the control tagging protein”. I would assume the difference is still highly significant when tested.

To clarify this confusion, we rewrote most of that paragraph to more precisely describe our results and point (lines 155-165).

Our point is to demonstrate that the mature CAL1 protein (without the secretion signal peptide at its N-terminus) can bind Cd at the cytosolic pH (7.5) and the apoplastic pH (5.5), and pH variation within this range had little effect on the Cd binding capacity (Fig. 4a). Although the full length CAL1 still binds Cd slightly more than the control tagging protein TF at pH 5.5, this binding capacity has no physiological meaning because the full length CAL1 would not be present in the apoplast (pH 5.5) in the real world.

b. The bars of new Fig. 4a resemble those of the previous Fig. 4a even though the assay conditions were changed quite a bit. However, if I am not mistaken the molar ratios calculated from previous Fig. 4a are about factor 10 different from the ratios shown in new Fig. 4a.

We confirm that the data present in new Fig. 4a is correct. We are puzzled why the reviewer got such a surprising value. It is possible that misunderstanding occurred to the reviewer: the CAL1 protein (5KD) was fused to the tag protein TF, so the molecular weight is ~60KD, not 5KD, which consequently would translate to a difference of about 10 fold, as the reviewer estimated.

c. the oxidation/reduction question is disregarded even though this should have a major influence on metal binding as can be expected for a protein with disulfide bridges and as shown, for example, with a human defensin in Zhang et al ACS Chem Biol 2013.

To address the reviewer's concern, we have added more sentences and one reference to better describe our point (lines 171-173).

We agree with the reviewer's point and actually have made it clear that this oxidation/reduction mechanism is common to those cysteine-rich proteins/peptides (providing thiols), such as phytochelatins, and also defensins as exemplified here by the reviewer. However, this mechanism is more about how exactly defensin binds ion, which has been answered by other groups as the reviewer noticed and is therefore not a focus of our study. Our study addresses the issue of whether the CAL1:Cd complex is secreted to the apoplast, which has not been reported before.

--Comment 6 - I do not see why it is not possible to derive an estimate of CAL1 protein level when the fusion protein can be detected in xylem sap with an anti-RFP antibody. The CAL1/Cd stoichiometry in the xylem sap is a crucial piece of information. As a side note: the apparent lack of a change in fusion protein levels in xylem sap in Cd treated plants (Fig. 4d) is somewhat inconsistent with the transcript data for roots.

To address the reviewer's concern (*originally, the reviewer proposed determination of the CAL1/Cd stoichiometry in xylem sap to give an estimation of how much CAL1 contributes to Cd translocation*), we have added more sentences to better describe our results and to give such an estimation using our mutant data (*lines 238-241, 155-165, 171-173*).

It is possible but very difficult to directly determine the CAL1/Cd stoichiometry in xylem sap, because doing so requires purification of the fusion protein (not necessary for Western blot), while the tag we used is mRFP, not His or other regular tags. The kit RFP-Trap[®]_A could be used to purify the CAL1:mRFP fusion, but it is very inefficient and expensive, thus we failed in several attempts.

Fortunately, without the CAL1/Cd stoichiometry in xylem sap, we are still able to estimate the contribution by CAL1 to Cd translocation using the *cal1* mutant. By comparing the Cd concentration changes in the

xylem sap from the *cal1* mutant with that from the wild-type, we estimate ~20% contribution by CAL1 to the Cd translocation rate. More sentences were added to describe this point (lines 238-241).

Please note we did measure the CAL1/Cd stoichiometry in *in vitro* assays using recombinant mature CAL1 proteins, which was ~1.2. We also showed that this stoichiometry was not affected by pH between 5.5 and 7.5 (Fig. 4a). In addition, we showed that CAL1 did not bind significant amounts of Ca, Mn or Zn (Supplementary Fig. 12). Furthermore, using site-directed mutagenesis of the cysteine residues, we found that only 3 of the 8 Cys residues in CAL1 significantly affect the Cd binding ability (Supplementary Fig. 11), suggesting that Cd is possibly coordinated to 3 thiol groups in CAL1, which would give rise to a very stable complex. Taken together, our results suggest that CAL1 most probably binds Cd in a 1:1 stoichiometry forming a Cd: 3(SH-) complex, similar to the Cd:phytochelatin complexes. The binding of CAL1 is highly specific to Cd and unaffected by pH or competing ions such as Ca, Zn and Mn. **Therefore, it is reasonable to predict that the CAL1/Cd stoichiometry in the xylem sap would be the same as in our *in vitro* assays.** We also added some words to better describe this result (lines 171-173).

In terms of the relevance between the protein and transcript levels, it is frequently observed that their levels do not necessarily show positive

correlation. We actually pointed this out in our previous version and proposed that CAL1 might have other unknown functions yet to be identified (*lines 204-207*).

--Comment 11 - Regarding my comment on variations in Cd levels across experiments: between Figs. 2f and 2g there is a difference of 10fold between the leaf values for two rice accessions; in another comparison TN1 seedlings accumulate around 20 µg/g Cd when exposed to 10 µM Cd for 7 days according to Fig. 1c while in Fig. 2f the value is around 100. These differences cannot be explained by batch-to-batch variation in my view. Rather, they point to possible technical problems.

Please note that TN1-CK genetic background in Fig 2f represents the Cd **over-accumulation** accession TN1, whereas CJ06-CK in Fig 2g represents the Cd **under-accumulation** accession CJ06. So it is reasonable that the Cd levels would be very different between Fig. 2f & 2g.

In terms of the comparison between Fig. 1c and Fig. 2f, we thank the reviewer for pointing this out, but we have to mention that these plants have different genetic background, one is wild type TN1 (*Fig 1f*), while the other one is TN1-CK, which as we described in the figure legend, is the transgenic negative control, meaning these plants were exposed to

callus induction and genetic transformation, so their genetic background already changed, which typically also contributes to phenotypic variation.

My lab is skilled in Ionomics, and decades of experience showed that batch-to-batch variation could be pretty large, especially for the quantitative ion accumulation. **However, this variation does not affect our conclusion, as we always compare the data in the same batch, or normalize the data to their own control before comparisons with other batches.** This strategy is widely accepted by scientists in this field.

In summary, we thank all the reviewers for their comments, which greatly enhanced our manuscript. We have also addressed the remaining concerns by reviewer #3 with new data and more revisions. The story reported here is novel and quite complex as large amounts of data are presented to raise multiple lines of evidence, thus the reviewers' comments have helped essentially to make our story concise and easier to be understood.

REVIEWERS' COMMENTS:

Reviewer #3 (Remarks to the Author):

The authors further improved the manuscript by adding new data and by clarifying some of their statements. Here are some specific comments to the points raised by the authors in their appeal:

Comment 2

What I meant is detection in the apoplast, not in the xylem exudates as presented by the authors. This could be done, for example, by obtaining apoplastic washing fluid from leaves.

Comment 3

It is of course asking a lot when insisting on more suitable transgenic controls. While I would hold up my view that plants expressing a non-secreted form of CAL1 would make the data much stronger I do appreciate the extra evidence suggesting presence of CAL1 in guttation fluid.

Comment 4

I acknowledge that my calculation was based on the molecular weight of about 5 kDa and not on the molecular weight of the fusion protein. Thus, the stoichiometry I derived was wrong.

Comment 6

I do not fully understand why it is necessary to purify the fusion protein for quantification. One could derive an estimate from Western blots when loading known amounts of an RFP fusion protein in a dilution series.

Comment 11

My point was that the differences in Cd levels far exceed the differences observed for the two accessions. However, I agree with the statement that what matters most is the comparison within experiments.

We thank the reviewer for the comments and suggestions regarding our manuscript entitled "Defensin like protein-driven cadmium efflux and allocation in rice"(NCOMMS-17-04721C). We have revised our manuscript accordingly to fully address the remaining concerns, which are highlighted in the main text of our manuscript.

Point-to-point response to reviewers

Reviewer #3:

-- *“Comment 2 - What I meant is detection in the apoplast, not in the xylem exudates as presented by the authors. This could be done, for example, by obtaining apoplastic washing fluid from leaves.”*

We are glad that with the guttation assay, the xylem data is now not a concern anymore. To clarify the remaining confusion, we improved the text (**lines 200-203**) to more accurately describe our point.

The original comment by the reviewer is *“Secretion should be further supported by independent evidence.....”*, and he/she suggested LC-MS detection of the CAL1 protein in the apoplast. We have addressed this question by providing two independent lines of evidence for the presence of CAL1 protein in the apoplast by collecting and analyzing the xylem sap and guttation sap, both representing the fluid from the apoplast. Although there is a possibility that xylem sap might be contaminated by

the symplastic content of the cut cells, guttation fluid was collected from the leaves of intact plants under natural conditions and was free of any symplastic contamination. The reviewer suggested collection of apoplastic washing fluid from leaves. However, just like collecting xylem sap, collecting the apoplastic washing fluid can also suffer from the contamination of partial cell lysis (Joosten, 2012, *Methods in Molecular Biology*, 835:603-). Therefore, we believe that collecting guttation fluid is a better method than collecting apoplastic washing fluid.

--Comment 3 - It is of course asking a lot when insisting on more suitable transgenic controls. While I would hold up my view that plants expressing a non-secreted form of CAL1 would make the data much stronger I do appreciate the extra evidence suggesting presence of CAL1 in guttation fluid.

We are glad that the reviewer agreed that the guttation fluid data provide the extra evidence for the presence of CAL1 in the apoplast.

--Comment 4 - I acknowledge that my calculation was based on the molecular weight of about 5 kDa and not on the molecular weight of the fusion protein. Thus, the stoichiometry I derived was wrong.

We are glad that the confusion is clarified.

--Comment 6 - I do not fully understand why it is necessary to purify the fusion protein for quantification. One could derive an estimate from Western blots when loading known amounts of an RFP fusion protein in a dilution series.

The original concern by the reviewer was how much CAL1 contributes to Cd translocation, and the reviewer suggested answering that question by estimating the CAL1:Cd stoichiometry in the xylem sap. We have actually answered that question by directly comparing the different Cd translocation between the wild type and *call* mutant rice plants. We have also estimated the CAL1:Cd stoichiometry in the xylem sap based on the *in vitro* assays. We are happy to see that this strategy is acceptable to the reviewer and is not a concern anymore.

To the best of our knowledge, it is necessary to purify the fusion protein for quantification because otherwise we cannot load “known amounts” of the RFP fusion as suggested by the reviewer. It is worth pointing out that the CAL1:Cd stoichiometry *in vitro* was determined using a different fusion protein (TF-CAL1), which was designed to express in *E. coli* and could be easily purified using commercially available columns.

--Comment 11 - My point was that the differences in Cd levels far exceed the differences observed for the two accessions. However, I agree with the

statement that what matters most is the comparison within experiments.

We are happy that our revision has clarified the confusion and addressed the reviewer's concern.

In summary, we believe that we have addressed the remaining concerns by reviewer #3, and would like to thank him/her for the critical review and constructive comments, which have been very useful for us to revise and strengthen our manuscript.